# ON THE GEOMETRY OF GENERALIZATION AND MEMORIZATION IN DEEP NEURAL NETWORKS

Cory Stephenson[*,1], Suchismita Padhy[*,1], Abhinav Ganesh[1], Yue Hui[2],
Hanlin Tang[1] and SueYeon Chung[+,3]

[1]Intel Labs, [2]Stanford University, [3]Columbia University,
*{cory.stephenson,suchismita.padhy,abhinav.ganesh,}@intel.com, yueh@stanford.edu,
hanlin.tang@intel.com, sueyeon.chung@columbia.edu*

## ABSTRACT

Understanding how large neural networks avoid memorizing training data is key to explaining their high generalization performance. To examine the structure of when and where memorization occurs in a deep network, we use a recently developed replica-based mean field theoretic geometric analysis method. We find that all layers preferentially learn from examples which share features, and link this behavior to generalization performance. Memorization predominately occurs in the deeper layers, due to decreasing object manifolds' radius and dimension, whereas early layers are minimally affected. This predicts that generalization can be restored by reverting the final few layer weights to earlier epochs before significant memorization occurred, which is confirmed by the experiments. Additionally, by studying generalization under different model sizes, we reveal the connection between the double descent phenomenon and the underlying model geometry. Finally, analytical analysis shows that networks avoid memorization early in training because close to initialization, the gradient contribution from permuted examples are small. These findings provide quantitative evidence for the structure of memorization across layers of a deep neural network, the drivers for such structure, and its connection to manifold geometric properties.

## 1 INTRODUCTION

Deep neural networks have many more learnable parameters than training examples, and could simply memorize the data instead of converging to a generalizable solution (Novak et al., 2018). Moreover, standard regularization methods are insufficient to eliminate memorization of random labels, and network complexity measures fail to account for the generalizability of large neural networks (Zhang et al., 2016; Neyshabur et al., 2014). Yet, even though memorizing solutions exist, they are rarely learned in practice by neural networks (Rolnick et al., 2017).

Recent work have shown that a combination of architecture and stochastic gradient descent implicitly bias the training dynamics towards generalizable solutions (Hardt et al., 2016; Soudry et al., 2018; Brutzkus et al., 2017; Li and Liang, 2018; Saxe et al., 2013; Lampinen and Ganguli, 2018). However, these claims study either linear networks or two-layer non-linear networks. For deep neural networks, open questions remain on the structure of memorization, such as where and when in the layers of the network is memorization occurring (e.g. evenly across all layers, gradually increasing with depth, or concentrated in early or late layers), and what are the drivers of this structure.

Analytical tools for linear networks such as eigenvalue decomposition cannot be directly applied to non-linear networks, so here we employ a recently developed geometric probe (Chung et al., 2018; Stephenson et al., 2019), based on replica mean field theory from statistical physics, to analyze the training dynamics and resulting structure of memorization. The probe measures not just the layer capacity, but also geometric properties of the object manifolds, explicitly linked by the theory.

We find that deep neural networks ignore randomly labeled data in the early layers and epochs, instead learning generalizing features. Memorization occurs abruptly with depth in the final layers, caused by

---

*: Equal contribution, +: Correspondence.

decreasing manifold radius and dimension, whereas early layers are minimally affected. Notably, this structure does not arise due to gradients vanishing with depth. Instead, analytical analysis show that near initialization, the gradients from noise examples contribute minimally to the total gradient, and that networks are able to ignore noise due to the existence of 'shared features' consisting of linear features shared by objects of the same class. Of practical consequence, generalization can then be re-gained by rolling back the parameters of the final layers of the network to an earlier epoch before the structural signatures of memorization occur.

Moreover, the 'double descent' phenomenon, where a model's generalization performance initially decreases with model size before increasing, is linked to the non-monotonic dimensionality expansion of the object manifolds, as measured by the geometric probe. The manifold dimensionality also undergoes double descent, whereas other geometric measures, such as radius and center correlation are monotonic with model size.

Our analysis reveals the structure of memorization in deep networks, and demonstrate the importance of measuring manifold geometric properties in tracing the effect of learning on neural networks.

## 2 RELATED WORK

By demonstrating that deep neural networks can easily fit random labels with standard training procedures, Zhang et al. (2016) showed that standard regularization strategies are not enough to prevent memorization. Since then, several works have explored the problem experimentally. Notably Arpit et al. (2017) examined the behavior of the network as a single unit when trained on random data, and observed that the training dynamics were qualitatively different when the network was trained on real data vs. noise. They observed experimentally that networks learn from 'simple' patterns in the data first when trained with gradient descent, but do not give a rigorous explanation of this effect.

In the case of deep linear models trained with mean squared error, more is known about the interplay between dynamics and generalization (Saxe et al., 2013). While these models are linear, the training dynamics are not, and interestingly, these networks preferentially learn large singular values of the input-output correlation matrix first. During training, the dynamics act like a singular value detection wave (Lampinen and Ganguli, 2018), and so memorization of noisy modes happens late in training.

Experimental works have explored the training dynamics using variants of Canonical Correlation Analysis (Raghu et al., 2017; Morcos et al., 2018), revealing different rates of learning across layers, and differences between networks that memorize or generalize (Morcos et al., 2018). Using Centered Kernel Alignment, Kornblith et al. (2019) find that networks differ from each other the most in later layers. However, as these metrics measure similarity, experiments are limited to comparing different networks, offering limited insight into specific network instances. As noted in (Wang et al., 2018), the similarity of networks trained on the same data with different initialization may be surprisingly low.

Our work builds on this line of research by using a direct, rather than comparative, theory-based measure of the representation geometry (Chung et al., 2016; Cohen et al., 2019) to probe the layerwise dynamics as learning gives way to memorization. With the direct theory-based measure, we can probe individual networks over the course of training, rather than comparing families of networks.

## 3 EXPERIMENTAL SETUP

As described in (Arpit et al., 2017), we adopt the view of memorization as "the behavior exhibited by DNNs trained on noise." We induce memorization by randomly permuting the labels for a fraction $\epsilon$ of the dataset. To fit these examples, a DNN must use spurious signals in the input to memorize the 'correct' random label. We train our models for either $1,000$ epochs, or until they achieve $> 99\%$ accuracy on the training set, implying that randomly labeled examples have been memorized. We do not use weight decay or any other regularization.

Once a model has been trained with partially randomized labels, we apply the recently developed replica-based mean-field theoretic manifold analysis technique (MFTMA hereafter) (Chung et al., 2018) to analyze the hidden representations learned by the model at different layers and epochs of training. This quantitative measure of the underlying manifold geometry learned by the DNN provides insight into how information about learned features is encoded in the network. This method

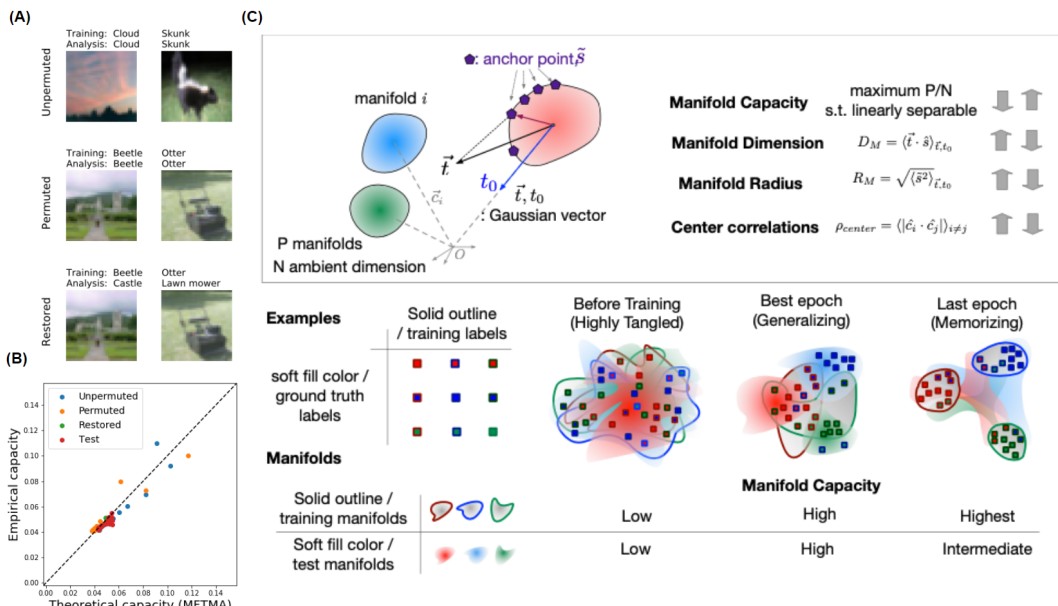

Figure 1: **Description of Manifold Analysis.** (**A**) Random samples of unpermuted, permuted, and restored examples showing the label used for training and our geometric analysis. Note that restored and permuted examples only differ on the label used for analysis. (**B**) Empirical manifold capacity matches theory. (**C**) (Top) MFTMA framework formally links manifold capacity, a measure object manifolds' linear separability, to geometric properties. Reduced dimension, radius, center correlations result in larger manifold capacity and vice versa. (Bottom) Hypothesis of how network training affects geometry and separability of object manifolds in the presence of label noise.

has been applied to pre-trained models (Cohen et al., 2019), whereas here we study the training dynamics in the memorization regime.

## 3.1 DATASETS AND MODELS

We carried out experiments on a variety of neural network architectures and datasets. For CIFAR-100 (Krizhevsky et al., 2009), we trained AlexNet (Krizhevsky et al., 2014), VGG-16 (Simonyan and Zisserman, 2014) and ResNet-18 (He et al., 2016) models, each slightly modified for use with the dataset. We also used Tiny ImageNet dataset[1], a reduced version of ImageNet (Deng et al., 2009) with ResNet-18. Training methods similar to those used for the CIFAR-100 dataset did not result in complete memorization on Tiny ImageNet, owing to the larger and more complex dataset, and so we show results on partial memorization after training for 1,000 epochs. Due to space constraints, we present results for VGG-16 trained on CIFAR-100. Other architectures and datasets have similar findings, as detailed in the Appendix.

Our networks are trained on a dataset with some fraction $\epsilon$ of its labels randomly permuted. To analyze the behavior of the trained models, we define four subsets of examples as follows:

**Unpermuted examples**: The set of labeled examples from the training set which did not have their labels permuted. That is, the networks were trained on these examples using the correct label. In the case of $\epsilon = 100\%$ permuted data, this set is empty, as we do not count examples with labels randomly assigned to the correct class.

**Permuted examples**: The set of labeled examples from the training set which had their labels randomly permuted. The networks were trained to classify these examples according to their random labels. By design, there is some chance that the random assignment results in the label remaining unchanged, however these examples are still in the set of the permuted examples. The motivation for this choice can be found in the Appendix.

**Restored examples**: These examples are the same as the permuted examples, except when carrying out analysis we restored the correct, unpermuted labels. For example, for a permuted example with

---

[1]This dataset is from the Tiny ImageNet Challenge: `https://tiny-imagenet.herokuapp.com/`

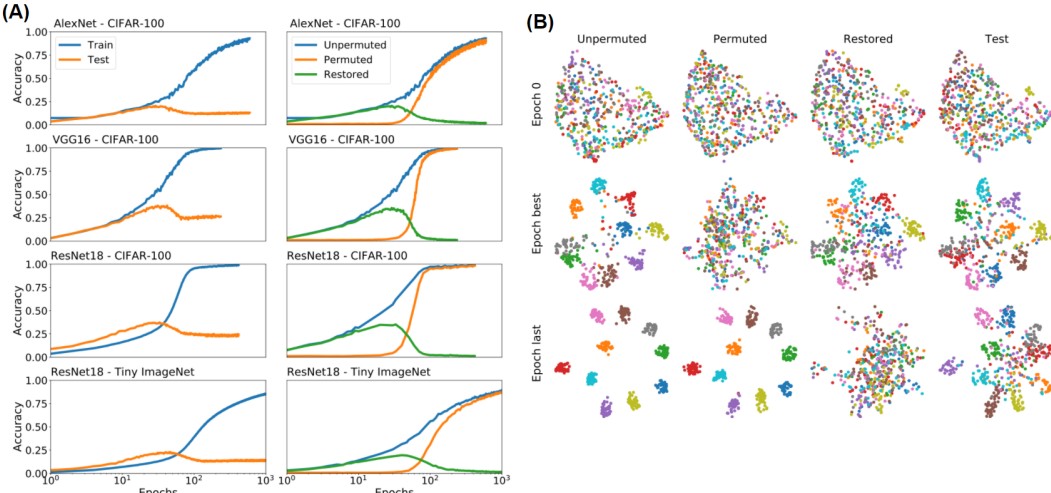

Figure 2: **Generalization and memorization as seen by accuracy and visualization. A** (left): Training and test accuracy curves for all models and datasets ($\epsilon = 50\%$). **A** (right): Accuracy on unpermuted, permuted, and restored training examples. **B**: UMAP visualization of the final layer features in VGG16 with $\epsilon = 50\%$ at initialization, best epoch, and final epoch. At the best epoch, unpermuted, restored, and test manifolds show similar structure, implying that the network has learned the features shared between the two datasets. First 10 classes of CIFAR100 are included.

an image of class 'cat' that was trained with label 'dog', the corresponding restored example would be the same image with the label 'cat' restored.

A sampling of unpermuted, permuted, and restored examples is shown in Fig. 1A, with the denoted labels used for training and analysis. Finally, we define test examples:

**Test examples**: The set of labeled examples from a holdout set not seen during training. All analyses on test examples are done using the correct label.

## 3.2 REPLICA-BASED MEAN FIELD THEORY MANIFOLD ANALYSIS

We apply the replica MFTMA analysis (Chung et al., 2018; Cohen et al., 2019; Stephenson et al., 2019) which formally connects object category manifolds' linear separability with the underlying geometric properties, such as manifold dimension, radius, and correlations.

A key feature in object manifold capacity theory is that it generalises the notion of perceptron shattering capacity from discrete points to class manifolds. Given $P$ object manifolds defined by the clouds of feature vectors in $N$ feature dimension, where each manifold is assigned random binary labels where points within the same manifold have an identical label, rendering a total of $2^P$ manifold dichotomies. The manifold capacity, $\alpha_c = P/N$ is defined by the critical number of object manifolds where most of the manifold dichotomies can be shattered (separated) by a linear hyperplane. This is closely related to Cover's function counting theorem, except in this theory, the units for counting $P$ for capacity are category manifolds, rather than discrete point patterns. The manifold capacity, therefore, measures the linear separability of categories in the representation. Surprisingly, it's been reported that the object manifold capacity for general point clouds can be formulated in a similar manner to the closed-form manifold capacity of $D$-dimensional balls of radius $R$ in random orientations (Chung et al., 2016), providing the closed form expressions for effective dimension $D_M$ and effective radius $R_M$ for the convex hulls of general point clouds in random orientation. More recently, the theory of manifolds has been refined to incorporate correlation structure in the manifold locations, allowing for the estimation of manifold capacities in real data with correlations (Cohen et al., 2019). This analysis, based on the series of theoretical developments mentioned above, formally links the linear separability of category class manifolds (manifold capacity, $\alpha_M$) with the class manifolds' geometric properties (manifold dimension, $D_M$, manifold radius, $R_M$, and the correlations between the manifold centers, $\rho_{center}$). The descriptions of the manifold capacity and the geometrical properties are summarized below (for more details, see Appendix).

**Manifold Capacity** $\alpha_M = P/N$ is a measure of the decodable class information in a given data, defined by critical number of object manifolds (P) that can be linearly classified with random dichotomy labels for each manifold, in feature dimension N. Large $\alpha_M$ implies well-separated manifolds in feature space. Using replica mean field theory, this quantity can be predicted (Chung et al., 2018) by object manifolds' dimension, radius, and their center correlation structure provided below (see illustrations in Fig. 1C and Appendix). Note that the manifold capacity can also be computed empirically (*Empirical capacity*, hereafter), by doing a bisection search to find the critical number of features N such that the fraction of linearly separable random manifold dichotomies is close to 1/2. We show that theoretical predictions match empirical capacity with our data in Fig. 1B.

**Manifold Dimension** $D_M$ captures the dimensions of the set of *anchor points*, which are representative support vectors that determine the optimal hyperplane separating the binary dichotomy of manifolds. These anchor points are estimated from the guiding Gaussian vectors (Fig. 1C and Appendix), and the position of the anchor point changes as orientations of other manifolds are varied. The average of the dot product between anchor points and the Gaussian vector defines the effective manifold dimension of given point clouds, and it reflects the dimensionality of embedded anchor points in the optimal margin hyperplane. A small dimension implies that the anchor points occupy a low dimensional subspace.

**Manifold Radius** $R_M$ is an average norm of the anchor points in the manifold subspace, capturing the manifold size seen by the decision boundary. Small $R_M$ implies tightly grouped anchor points.

**Center Correlation** $\rho_{center}$ measures the average of absolute values of pairwise correlations between various centers. A small correlation indicates that, on average, manifolds lie along different directions in feature space.

In the rest of the text, we will refer to the quantities $\alpha_M$, $D_M$, $R_M$, and $\rho_M$ as the Manifold Geometry Metrics (MGMs). Small values for manifold radius, dimension and center correlation have been shown to result in higher values for manifold capacity (Chung et al., 2018; Cohen et al., 2019), and so represent a more favorable geometry for classification (Fig. 1C). For all experiments, we evaluate these four quantities using $P = 100$ classes, with $M = 50$ randomly drawn examples per class.

We define unpermuted, permuted, restored, and test manifolds for each class at each layer of the network. Given a neural network $N(x)$, we extract $N_l(x)$, the activation vector at layer $l$ generated on input $x$. We then define a manifold by the point cloud of activation vectors generated by inputs of the same class. This results in $P$ manifolds, one for each class. The set of unpermuted manifolds is then obtained by using the corresponding manifolds with examples and labels that were unpermuted. The set of permuted, restored, and test manifolds are defined similarly. We run the MFTMA on each set of manifolds following the procedures in (Chung et al., 2018; Cohen et al., 2019; Stephenson et al., 2019). All results are conducted with 10 random seeds, which govern the network initialization, label permutation, and MFMTA analysis initial conditions.

## 4 RESULTS

The networks we tested demonstrate memorization of the training dataset, with the eventual decrease of the test accuracy (Fig.2A, left orange curves) and near 100% training accuracy (blue curves). Training with the original dataset also demonstrates this behavior, albeit less exaggerated. More epochs are required to fit the permuted examples (Fig. 2A, right, orange curves), consistent with (Zhang et al., 2016). However, these examples quickly converge once they begin, reminiscent of the singular value detection wave observed in deep linear networks (Saxe et al., 2013; Lampinen and Ganguli, 2018). Importantly, before the permuted examples are learned, many are actually classified as their *correct* label, as shown by the accuracy increase in the restored examples (right, green curves). This is despite the model being trained to emit the wrong randomized label for the restored inputs. Intuitively, restored examples are classified correctly because the networks have learned features that generalize beyond the set of unpermuted (correctly labeled) training examples.

We visualize this effect with a UMAP (McInnes et al., 2018) of the final layer features (Fig. 2B). At the best generalizing epoch, the unpermuted, restored, and test examples are separable while permuted examples are mixed. By the last epoch, however, separability is high for unpermuted and permuted examples, indicating memorization, while restored examples are no longer separable.

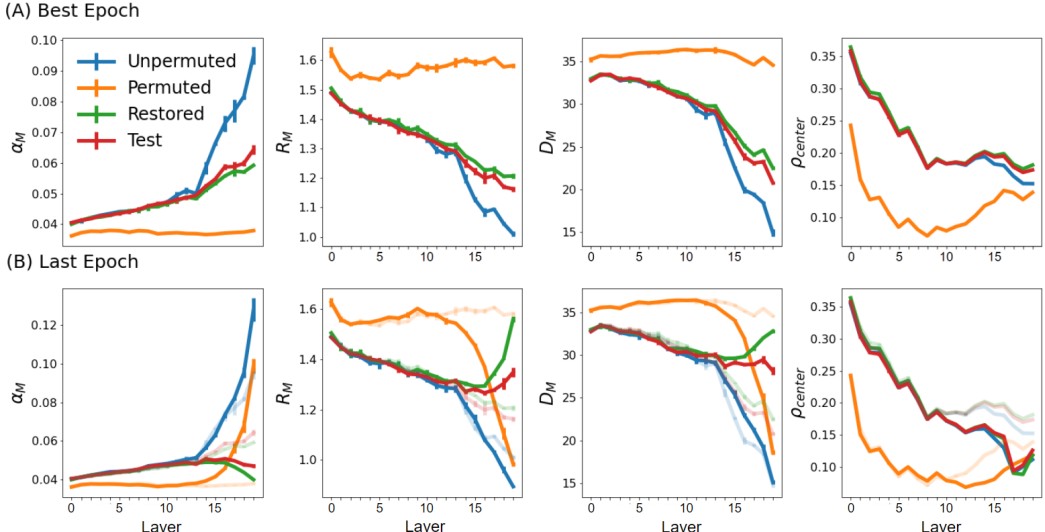

Figure 3: **MFT Geometry of generalization and memorization on VGG16. A**: Best Epoch. Geometry of restored and test manifolds show no significant differences, and permuted manifolds show no evidence of learning. Error bars represent 99% CI of the mean. **B**: Final epoch, after significant memorization has occurred. Restored and test manifolds show similar trends, and only differ from unpermuted manifolds after layer 14. Faded lines show the results at best epoch for ease of comparison.

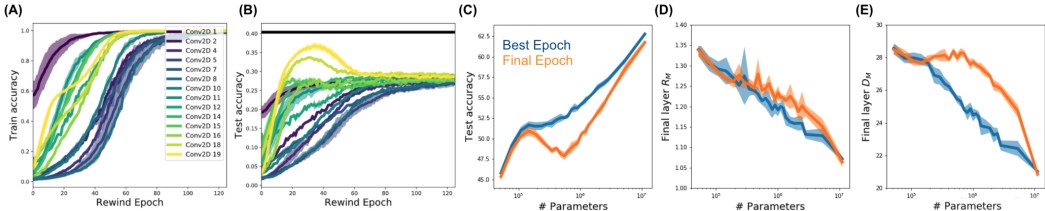

Figure 4: **(A-B) Rewinding training**: Accuracy on the train set (A), and the test set (B) for VGG-16 trained for 1000 epochs with an individual layer's parameters rolled back to an earlier epoch (rewind epoch). The final three linear layers of VGG-16 are unchanged throughout. Horizontal line indicates the peak generalization performance from early stopping. **(C-E) Double descent phenomenon**, with increasing model parameters, at the best epoch (blue) and final epoch (orange). The region is linked to increased manifold dimensionality (E), with unchanged radius and center correlation.

**Generalizing features are learned in early layers and epochs:** MFTMA provides a fine-grained view into how each layer's features encode class information by measuring the manifold geometry metrics (MGMs). Figure 3 shows the MGMs computed on VGG-16 trained on CIFAR-100. Trends are similar across other architectures, permutation fractions and datasets, as shown in the Appendix.

At peak generalization (the best epoch, with highest test set accuracy - Fig. 3A), the early layers exhibit no difference between the manifolds (excluding the permuted manifold). For deeper layers, the manifolds increase in capacity $\alpha_M$ indicating that their features contain more information about the *correct* class label as depth increases. Importantly, this even holds for the restored manifolds (green curve), despite the model being explicitly trained to produce a random label. Similarly, the manifold radius $R_M$ and dimension $D_M$ decreases for the restored manifolds. This shows quantitatively that in the early epochs of training (up to the point of early stopping) the features learned by all layers, but especially the early layers, of the model generalize across many examples of a class. In contrast, the manifold capacity for permuted manifolds remains near the theoretical lower bound[2], as the network has not yet memorized the randomly selected labels.

---

[2]The theoretical lower bound in this case is $\alpha_M = \frac{2}{M} = 0.04$ from Cover's theorem (Cover, 1965)

**Memorization occurs abruptly in later layers and epochs:**  At the end of training, after memorization, a different set of trends emerges, as shown in Fig. 3B. The unpermuted manifolds (blue curve) behavior similarly to the best epoch (blue, shaded). The permuted manifolds (orange) however show a dramatically different trend; the manifold capacity is increased significantly in later layers ($Conv_{16}$ and onwards), as the network has learned to group these examples by their random labels. Similarly, the manifold radius and dimension for permuted examples decrease significantly. In these same later layers, however, the restored manifolds (green) now have decreased capacity compared to the best epoch. While the restored manifolds were initially untangled by the early layers, the later layers have learned to tangle them again as the permuted examples are memorized.

The dramatic trend reversals above are not observed in the early layers. Even after the permuted examples have been memorized, the feature geometry has not changed significantly, implying that the learned features in early layers are still fairly general, with examples grouped according to their true labels even if their training label was randomly assigned.

**Undoing memorization by rewinding individual layers:**  If memorization occurs mostly in later layers, the effects of memorization should be undone by selectively modifying only the weights of the final layers. We carried out a rewinding experiment inspired by Zhang et al. (2019). Using the final model trained for 1,000 epochs, we selectively rewind the parameters of a single layer to their values at an earlier epoch (the rewind epoch $E$). Whereas Zhang et al. (2019) only examine $E = 0$, we rewind to various $E$ throughout training to probe the dynamics. Indeed, as shown in Fig. 4B, simply rewinding the parameters of the final *convolutional* layer (yellow curve) to an earlier epoch reverses the memorization effect, yielding a model that is >90% as accurate as the best early-stopped model. Note that the final three linear layers are not being rewound. This is similar to the freeze training method (Raghu et al., 2017), which selectively freezes *early* layers, but here we do not modify the training procedure, and find rather that *later* layers must be rewound. Furthermore, note that early layers (blue curves) monotonically approach their final performance, whereas later layers (green-yellow) initially learn weights that generalize before reaching the memorizing state.

Prior work (Raghu et al., 2017; Morcos et al., 2018) show that early *representations* stabilize first, which is consistent with our MFTMA analysis of representations (Fig. 3). However, the behavior of the network *weights* exhibit the opposite trend; early layer weights (blue curves) take longer to stabilize compared to deeper layer weights (green-yellow curves) (Fig. 4A).

**Double descent phenomenon in manifold dimensions:**  The *double descent* phenomenon is where the generalization performance gets worse before getting better as the model size gets larger. To check the representational correlates underlying this phenomenon, we varied the parameter count of a ResNet-18 network by scaling the layer width by a common factor, and trained on Cifar100 with 10% label noise (similar to (Nakkiran et al., 2019)). As expected, the model accuracy exhibits double descent (Fig. 4C). What are neural representations underlying this phenomenon? Our geometric analysis on test data shows that the object manifolds' dimensionality at the final epoch also first decreases, then increases, and decreases again with larger model size, whereas other geometric properties have monotonic relationships similar to the best epoch (Fig. 4D-E).

Manifolds dimensions are linked to the dimensions embedded to the linear classifying hyperplanes (Chung et al., 2018), and higher object manifold dimension renders unfavorable geometry for linear classification, explaining the lower test accuracy. Larger models with higher feature dimensions allow a higher upper bound for object manifolds' dimension. However, large feature dimensions also allow more room for object manifolds without needing to be packed in the classification hyperplane, which can result in lower manifold dimensions. These two factors compete for object manifolds' dimensionality, a property closely linked to their separability. Our geometric analysis reveals that the dimensionality of data manifolds associated with each object class also shows a double descent phenomenon, recasting the problem into that of representations. Investigation of the underlying mechanisms are an important open problem for future work.

**Gradient descent dynamics initially ignores permuted examples:** A tempting explanation for the lack of memorization and slower training dynamics observed in the early layers is vanishing gradients (Glorot and Bengio, 2010) in early compared to later layers. However, as shown in Fig. 5 (A), the gradients do not vanish in the early layers, and are not small until well after memorization.

**Figure 5: Gradient analysis. (A-C):** $\ell^2$ norm of the gradient of the loss over the different layers and epochs of training for VGG16, computed on (A) all samples, (B) unpermuted samples only, and (C) permuted samples only. **(D):** Ratio of gradients. For the label dependent part ($dep$) of the gradient, the gradients averaged from the unpermuted samples are much larger than the gradient from permuted samples (green line). For the independent part, the two components are roughly equal (Red).

Additionally, the unpermuted and permuted examples contribute similarly to the total gradient during the last phase of training as shown in Fig. 5 (B, C).

While the gradient magnitudes cannot explain our observations, a more careful analysis of the gradient descent dynamics reveals their origin; near initialization, the gradient from the permuted examples are minimal, with learning dominated by gradients from the unpermuted examples. For tractability, we analyze gradient descent instead of SGD, consistent with other work (Lampinen and Ganguli, 2018; Advani and Saxe, 2017). For the large batch sizes and small learning rates in our experiments, our experimental findings are reflected in the much simpler case of gradient descent. Consider a neural network $N(x, \theta)$ with input $x$ and parameters $\theta$. We obtain the model $P_M$ by applying a softmax, e.g. $P_M(c|x, \theta) = \mathcal{Z}^{-1} e^{N_c(x, \theta)}$ with $\mathcal{Z} = \sum_{c=1}^{P} e^{N_c(x, \theta)}$ for classes $c \in \{1, 2, ..., P\}$. When training the network to minimize the cross entropy cost function $\mathcal{L}$ with respect to the label distribution $P_L(c|x, \theta)$, the following claims hold true:

**Claim 1.** *The gradient of the cross entropy can be written as the sum of a label dependent and a label independent part: $\nabla_\theta \mathcal{L} = \nabla_\theta \mathcal{L}_{dep} + \nabla_\theta \mathcal{L}_{ind}$.*

We write the cross entropy loss and substitute the softmax model $P_M$ above to obtain the following decomposition, where $\langle \cdot \rangle_\mathcal{D}$ denotes the average over samples of the entire dataset,

$$\mathcal{L} = -\left\langle \sum_c P_L(c|x_i) \log(P_M(c|x)) \right\rangle_\mathcal{D} = -\left\langle \sum_c P_L(c|x_i) N_c(X) \right\rangle_\mathcal{D} + \left\langle \log \mathcal{Z} \right\rangle_\mathcal{D} \quad (1)$$

The first operand above depends on the labels $P_L$, and we denote as $\mathcal{L}_{dep}$. For the second operand, the labels $P_L$ were marginalized away, leaving only the partition function of the softmax, which is label independent ($\mathcal{L}_{ind}$). See the Appendix for more details.

**Claim 2.** *For a feed-forward network, the label dependent and label independent components of the gradient can be understood in terms of shared features.*

For the final layer $L$ of the network, we can write the gradients of both components as the weighted average of the activations, centered by the average activation (see Appendix for full derivation):

$$\frac{\partial \mathcal{L}_{dep}}{\partial w_{\alpha\beta}^L} = -\left\langle P_L(\alpha|x_i) \left( \phi_\beta^{L-1}(x_i) - \bar{\phi}_\beta^{L-1} \right) \right\rangle_\mathcal{D}$$
$$\frac{\partial \mathcal{L}_{ind}}{\partial w_{\alpha\beta}^L} = +\left\langle P_M(\alpha|x_i) \left( \phi_\beta^{L-1}(x_i) - \bar{\phi}_\beta^{L-1} \right) \right\rangle_\mathcal{D} \quad (2)$$

Here, $\alpha$ denotes the class and $\phi_\beta^{L-1}(x_i)$ is the activation of unit $\beta$ for the penultimate layer given input $x_i$. By inspection of Eq. 2, we see that to contribute to training, individual examples must generate an activation $\phi_\beta$ that differs significantly from the average activation $\bar{\phi}_\beta \equiv \langle \phi_\beta \rangle_\mathcal{D}$.

Furthermore, the label dependent part of the gradient is enhanced when examples of the same class cause similar patterns of activations relative to the global mean, and therefore add constructively

to the gradient during the $\langle \cdot \rangle_{\mathcal{D}}$ operation. Intuitively, this means many examples of a class must share the same attribute that $\phi_\beta^{L-1}$ codes. Conversely, if different examples of a class activate $\phi_\beta^{L-1}$ differently, the gradients add destructively. The situation for other layers is similar as shown in the Appendix, with an added minor complexity due to the jacobian of subsequent layers.

**Claim 3.** *Near initialization, the label dependent gradient vanishes for permuted examples, and the label independent term is small for both unpermuted and permuted examples.*

In standard initialization schemes (Glorot and Bengio, 2010), the network output distribution $P_M$ is near uniform at the start of training, implying $\partial \mathcal{L}_{ind} \approx 0$ as the two terms cancel in Eq. 2.

The label dependent part $\partial \mathcal{L}_{dep}$ has a qualitatively distinct behavior in the early epochs, as the training labels $P_L$ are not uniformly distributed. However, for the permuted examples additional simplification is possible. As Claim 2 shows, the label dependent gradient depends on the cross correlation between the centered features $\phi_\beta^{L-1}(x_i) - \bar{\phi}_\beta^{L-1}$ and the labels $P_L(c|x)$. Since permuted examples are constructed by random labeling, this cross correlation vanishes for large datasets. Thus, the dominant contribution to the learning comes from the correctly labeled examples. At random initialization, correctly labeled examples may still have shared features while the permuted examples do not. Previous work (Jacot et al., 2018; Lee et al., 2019) suggests that very wide networks remain near initialization where our analysis in Claim 3 holds, so we hypothesize that wider networks rely more on shared features, though this may not always be optimal for accuracy (Tsipras et al., 2019).

These claims are supported experimentally, as shown in Fig. 5D, which compares the relative sizes of the gradients throughout training. For the unpermuted examples (Fig. 5D), $\partial \mathcal{L}_{ind}$ is small compared to $\partial \mathcal{L}_{dep}$ initially, and equilibriate only later. For permuted examples (Fig. 5E), due to the vanishing cross-correlation mentioned above, instead $\partial \mathcal{L}_{dep}$ is comparatively smaller to $\partial \mathcal{L}_{ind}$, indicating that these examples do not share features at initialization. Consistent with the claims above, for the label dependent part of the gradient (Fig. 5F), the contribution from the unpermuted examples are initially significantly larger than from the permuted examples.

## 5 DISCUSSION

We provide quantitative evidence, based on representational geometry, of when and where memorization occurs throughout deep neural networks, and the drivers for such structure. The findings go beyond the comparative results (Raghu et al., 2017; Morcos et al., 2018; Kornblith et al., 2019) with direct probing of network instances, and provide empirical evidence and theoretical explanation to the qualitative findings of (Arpit et al., 2017). In addition, our observation linking double descent to increasing object manifold dimensionality highlights the value of geometric probes, recasting the phenomenon into that of feature representations, and opening a new direction for future study.

With standard initialization techniques, the approximate linearity of modern deep networks at the beginning of training (Lee et al., 2019) implies that the networks are sensitive to features that are approximately linear projections of the data. Since randomly labeled examples have few if any shared features, they make a minimal contribution to the gradient at the start of training. Therefore, the network learns primarily from the correct labels until the network leaves the linear regime. Our experimental results support this hypothesis; representations early in training are not impacted by permuted data, and such data have smaller gradient contribution than from the correctly labeled data. Our conclusions are consistent with prior reports on Coherent Gradients (Chatterjee, 2020).

We additionally observed several surprising phenomenon that warrant further investigation. The abrupt instead of gradual transition in memorization with layer depth (Fig. 3) is inconsistent with previous expectations (Morcos et al., 2018). The lack of gradient magnitude differences between early and late layers, even though they behave qualitatively different. The effect of double descent phenomenon on the measured manifold geometries and their competing underlying structural causes. The different generalization properties of early and later layers deep into training. These observations and future work are enabled by our more precise and rigorous investigation of the structure across layers of memorization in deep neural networks.

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

## A  SUPPLEMENTAL INFORMATION

### A.1  MEAN FIELD THEORETIC MANIFOLD ANALYSIS (MFTMA)

Here we provide a summary of replica-based Mean-Field Theoretic Manifold Analysis (MFTMA) used in this paper. First introduced in (Chung et al., 2018), this framework has been used to analyze internal representations of deep networks for visual (Cohen et al., 2019), speech (Stephenson et al., 2019) and natural language tasks (Mamou et al., 2020). MFTMA framework is a generalization of statistical mechanical theory of perceptron capacity for discrete points (Gardner, 1988) to manifolds. Traditional point perceptron capacity measures the maximum number of points in general position that can be linearly separated given the feature dimension and the ensemble of random dichotomy labels for points. Whereas the MFTMA framework measures the "manifold" capacity, defined as the maximum number of classes that can be linearly separated given the ensemble of random dichotomy labels for classes. MFTMA framework allows for the characterization of geometric properties of object manifolds in connection to their linear separability, as it has been shown that the object manifolds' properties can be used to predict the manifold classification capacity (Chung et al., 2018; Cohen et al., 2019; Stephenson et al., 2019; Mamou et al., 2020). As the framework formally connects the representational geometric properties and the manifold classification capacity, the measures from this framework are particularly useful in understanding how information content about object categories are embedded in the structure of the internal representations from the end-to-end trained systems (such as models from our paper).

### A.1.1 MANIFOLD CAPACITY

In a system where $P$ object manifolds are embedded in $N$-dimensional ambient feature space, *load* is defined as $P/N$. Given a fixed feature dimension $N$, large/small load implies that many/few object manifolds are embedded in the feature dimension. Consider a linear classification problem where positive and negative labels are assigned randomly to $P$ object manifolds (note that all the points within the same object class share the same label), and the problem is to find a linearly classifying hyperplane for these random manifold dichotomies. As each of the $P$ manifolds are allowed positive/negative labels, there are $2^P$ manifold dichotomies. Manifold capacity is defined as the critical load $\alpha_c = P/N$ such that above it, most of the dichotomies have linearly separating solution, and below it, most of the dichotomies do not have linearly separating solution. Intuitively, a system with a large manifold capacity has object manifolds that are well separated in the feature space, and a system with a small manifold capacity has object manifolds that are tangled in the feature space.

**Interpretation** Manifold capacity has multiple useful interpretations. First, as the manifold capacity is defined as the critical load for a linear classification task, it captures the linear separability of object manifolds. Second, the manifold capacity is defined as the *maximum* number of object manifolds that can be packed in the feature space such that they are linearly separable, it has a meaning of how many object manifolds can be "stored" in a given representation such that they are linearly separable. Third, as linear classifier is often thought of as a linear decoder, the manifold capacity captures the amount of linearly decodable object information per feature dimension embedded in the distributed representation.

**Mean-Field-Theory (MFT) Manifold Capacity** Manifold capacity, $\alpha_M = P/N$, can be estimated using the replica mean field formalism with the framework introduced by (Chung et al., 2018; Cohen et al., 2019). As mentioned in the main text, $\alpha_M$ is estimated as $\alpha_{MFT}$, for MFT manifold capacity, from the statistics of *anchor points* , $\tilde{s}$, a representative point for the points within a object manifold that contributes to a linear classification solution.

The general form of the MFT manifold capacity, exact in the thermodynamic limit, has been shown (Chung et al., 2018; Cohen et al., 2019) to be:

$$\alpha_{MFT}^{-1} = \left\langle \frac{\left[ t_0 + \vec{t} \cdot \tilde{s}(\vec{t}) \right]_+^2}{1 + \left\| \tilde{s}(\vec{t}) \right\|^2} \right\rangle_{\vec{t}, t_0}$$

where $\langle \ldots \rangle_{\vec{t}, t_0}$ is a mean over random $D$- and 1- dimensional Gaussian vectors $\vec{t}, t_0$ whose components are i.i.d. normally distributed $t_i \sim \mathcal{N}(0, 1)$.

This framework introduces the notion of *anchor points*, $\tilde{s}$, uniquely given by each $\vec{t}, t_0$, representing the variability introduced by all other object manifolds, in their arbitrary orientations. $\tilde{s}$ represents a weighted sum of support vectors contributing to the linearly separating hyperplane in KKT (Karush–Kuhn–Tucker) interpretation.

### A.1.2 MANIFOLD GEOMETRIC MEASURES

The statistics of these anchor points play a key role in estimating the manifold's geometric properties, as they are defined as: $R_M^2 = \left\langle \left\| \tilde{s}(\vec{T}) \right\|^2 \right\rangle_{\vec{T}}$ and $D_M = \left\langle \left( \vec{t} \cdot \hat{s}(\vec{T}) \right)^2 \right\rangle_{\vec{T}}$ where $s = \hat{\tilde{s}}/\|\tilde{s}\|$ is a unit vector in the direction of $\tilde{s}$, and $\vec{T} = (\vec{t}, t_0)$, which is a combined coordinate for manifold's embedded space, $\vec{t}$, and manifold's center direction $t_0$.

The manifold dimension measures the dimensionality of the projection of $\vec{t}$ on its unique anchor point $\tilde{s}$, capturing the dimensionality of the regions of the manifolds playing the role of support vectors. In other words, the manifold dimension is the dimensionality of the object manifolds realized by the linearly separating hyperplane.

If the object manifold centers are in random locations and orientations, the geometric properties predict the MFT manifold capacity (Chung et al., 2018), by $\alpha_{\text{MFT}} \approx \alpha_{\text{Ball}}(R_M, D_M)$ where, $\alpha_{\text{Ball}}^{-1}(R, D) = \int_{-\infty}^{R\sqrt{D}} Dt_0 \frac{(R\sqrt{D} - t_0)^2}{R^2 + 1}$ as defined in (Chung et al., 2016).

In real data, the manifolds have various correlations, hence the above formalism has been applied to the data projected into the null spaces of manifold centers, similar to the method proposed by (Cohen et al., 2019).

The validity of this method is shown various literature (Cohen et al., 2019; Stephenson et al., 2019; Mamou et al., 2020), where the good match between the MFT manifold capacity and a ground truth (empirical) capacity has been demonstrated (computed using a bisection search on the critical number of feature dimensions required in order to reach roughly 50% chance of having linearly separable solutions given a fixed number of manifolds and their geometries). Hence, the manifold capacity $\alpha_M$ refers to the mean-field approximation of the manifold

capacity, $\alpha_{MFT}$. For more details on the theoretical derivations and interpretations for the mean-field theoretic algorithm, see (Cohen et al., 2019)(Chung et al., 2018).

## A.2 DERIVATION OF CLAIMS

In this section, we provide a detailed derivation of the three claims made in the main text. A deep neural network with a $P$-class softmax applied to the output has an output distribution of the form

$$P_M(c|x) = \mathcal{Z}^{-1} e^{N_c(x)} \tag{3}$$

With $\mathcal{Z} = \sum_{k=1}^{P} e^{N_k(x)}$. Here, $P_M(c|x)$ is to be interpreted as the probability the model assigns to class $c$ given $x$, and $N_c(x)$ is the $c^{th}$ component of the network's output given the input $x$. Given training labels $P_L(c|x_i)$ (frequently one-hot, but not necessarily so) for each input $x_i$ in the training set $\mathcal{D}$, the network is trained to minimize the cross entropy loss

$$\mathcal{L} = -\left\langle \sum_{c=1}^{P} P_L(c|x) \log\left(P_M(c|x)\right) \right\rangle_{\mathcal{D}} \tag{4}$$

Here, $\langle \cdot \rangle_{\mathcal{D}} = \frac{1}{\mathcal{D}} \sum_{i=1}^{\mathcal{D}} \cdot$ is the uniform average over the samples in the dataset.

### A.2.1 DERIVATION OF CLAIM 1

Due to the normalized form of $P_M(c|x)$, the gradient of the cross entropy decouples into two parts:

$$\mathcal{L} = -\left\langle \sum_{c=1}^{P} P_L(c|x) N_c(x) \right\rangle_{\mathcal{D}} + \left\langle \log\left(\mathcal{Z}\right) \right\rangle_{\mathcal{D}} \tag{5}$$

Note that because $\mathcal{Z}$ is independent of the class label, the labels $P_L(c|x)$ sum away in the second term. Taking the gradient with respect to the parameters $\theta$ gives

$$\nabla_\theta \mathcal{L} = -\left\langle \sum_{c=1}^{P} P_L(c|x) \nabla_\theta N_c(x) \right\rangle_{\mathcal{D}} + \left\langle \nabla_\theta \log\left(\mathcal{Z}\right) \right\rangle_{\mathcal{D}} \tag{6}$$

Noting that the first term on the right hand side depends on the labels $P_L(c|x)$ while the second term does not, we can write

$$\nabla_\theta \mathcal{L} = \nabla_\theta \mathcal{L}_{dep} + \nabla_\theta \mathcal{L}_{ind} \tag{7}$$

Denoting that the gradient of the cross entropy with respect to the model parameters $\theta$ can be written as a sum of a label dependent part $\nabla_\theta \mathcal{L}_{dep}$ that depends on $P_L(c|x)$, and a label independent part $\nabla_\theta \mathcal{L}_{ind}$ that is completely independent of $P_L(c|x)$. The specific form of these two terms is dependent upon the parameterization of the model.

### A.2.2 DERIVATION OF CLAIM 2

Continuing from section A.2.1, we make use of a convenient identity that follows from the form of the softmax defined in Eq. 3:

$$\begin{aligned} \nabla_\theta \log\left(\mathcal{Z}\right) &= \mathcal{Z}^{-1} \sum_{k=1}^{P} \nabla_\theta e^{N_k(x)} \\ &= \mathcal{Z}^{-1} \sum_{k=1}^{P} e^{N_k(x)} \nabla_\theta N_k(x) \\ &= \sum_{k=1}^{P} P_M(k|x) \nabla_\theta N_k(x) \end{aligned} \tag{8}$$

Using this, the gradient of the cross entropy with respect to the network parameters $\theta$ can then be written as

$$\nabla_\theta \mathcal{L} = -\left\langle \sum_{c=1}^{P} P_L(c|x_i) \nabla_\theta N_c(x) \right\rangle_{\mathcal{D}} + \left\langle \sum_{c=1}^{P} P_M(c|x_i) \nabla_\theta N_c(x) \right\rangle_{\mathcal{D}} \tag{9}$$

After relabling the summation index $k$. If the network takes on a feedforward structure (omitting biases for clarity) we have

$$N_c(x) = W^L \phi^{L-1} \left( W^{L-1} \ldots \phi^1\left(W^1 x\right)\right) \tag{10}$$

Where $W^l$ and $\phi^l$ is the weight matrix and nonlinearity respectively at layer $l$. Via a straightforward application of the chain rule, the partial derivative of $N_c(x)$ w.r.t. the weight $w_{\alpha\beta}^l$, an element of the weight matrix at layer $l$, can be written as

$$\frac{\partial N_c(x)}{\partial w_{\alpha\beta}^l} = J_{c\alpha}^l(x)\phi_\beta^{l-1}(x) \equiv g_{c\alpha\beta}^l(x) \tag{11}$$

Here, $J_{c\alpha}^l(x)$ is the Jacobian of the network from layer $l$ to the final layer, and is a function of $x$ due to nonlinearity. $J_{c\alpha}^l(x)$ can be interpreted as the sensitivity of $N_c(x)$ due to changes in the pre-activation of feature $\alpha$ at layer $l$. The quantity $\phi_\beta^{l-1}(x)$ is an element $\beta$ of the activation vector at layer $l - 1$, and can be interpreted as the $\beta^{th}$ feature learned by the network at layer $l - 1$. The combination $g_{c\alpha\beta}^l(x)$ encodes how sensitive the network output at $c$ is to a small change in $w_{\alpha\beta}^l$ given the input $x$.

The partial derivative of the loss is then

$$\frac{\partial \mathcal{L}}{\partial w_{\alpha\beta}^l} = -\left\langle \sum_{c=1}^P P_L(c|x_i)g_{c\alpha\beta}^l(x) \right\rangle_\mathcal{D} + \left\langle \sum_{c=1}^P P_M(c|x_i)g_{c\alpha\beta}^l(x) \right\rangle_\mathcal{D} \tag{12}$$

Further insight can be gained by inserting a factor of $0 = \bar{g}_{c\alpha\beta}^l - \bar{g}_{c\alpha\beta}^l$ with

$$\bar{g}_{c\alpha\beta}^l \equiv \left\langle \frac{1}{C} \sum_{c=1}^P J_{c\alpha}^l(x)\phi_\beta^{l-1}(x) \right\rangle_\mathcal{D} \tag{13}$$

Which is the average partial derivative of $N_c(x)$ over all classes and all datapoints in the dataset. Further using the normalization of the label and model distributions, we can write $\bar{g}_{c\alpha\beta}^l = \sum_{c=1}^P P_L(c|x_i)\bar{g}_{c\alpha\beta}^l$ and $\bar{g}_{c\alpha\beta}^l = \sum_{c=1}^P P_M(c|x_i)\bar{g}_{c\alpha\beta}^l$. Then we can write

$$\frac{\partial \mathcal{L}}{\partial w_{\alpha\beta}^l} = \frac{\partial \mathcal{L}_{dep}}{\partial w_{\alpha\beta}^l} + \frac{\partial \mathcal{L}_{ind}}{\partial w_{\alpha\beta}^l} \tag{14}$$

Where we have introduced the label dependent part of the gradient $\frac{\partial \mathcal{L}_{dep}}{\partial w_{\alpha\beta}^l}$ and the label independent part of the gradient $\frac{\partial \mathcal{L}_{ind}}{\partial w_{\alpha\beta}^l}$ as introduced in section A.2.1. Using $\bar{g}_{c\alpha\beta}^l$, these can be written as

$$
\begin{aligned}
\frac{\partial \mathcal{L}_{dep}}{\partial w_{\alpha\beta}^l} &= -\left\langle \sum_{c=1}^P P_L(c|x_i)\left(g_{c\alpha\beta}^l(x) - \bar{g}_{c\alpha\beta}^l\right) \right\rangle_\mathcal{D} \\
\frac{\partial \mathcal{L}_{ind}}{\partial w_{\alpha\beta}^l} &= +\left\langle \sum_{c=1}^P P_M(c|x_i)\left(g_{c\alpha\beta}^l(x) - \bar{g}_{c\alpha\beta}^l\right) \right\rangle_\mathcal{D}
\end{aligned}
\tag{15}
$$

An important property for these two terms is that in this form, we have the special case of $\frac{\partial \mathcal{L}_{dep}}{\partial w_{\alpha\beta}^l} = 0$ if the training label distribution $P_L(c|x)$ is uniform, and similarly $\frac{\partial \mathcal{L}_{ind}}{\partial w_{\alpha\beta}^l} = 0$ if the network's output distribution $P_M(c|x)$ is uniform. This follows from the definition of $\bar{g}_{c\alpha\beta}^l$ as the average over points in the dataset with a uniform measure over classes.

For the special case of layer $L$, the final layer of the network, we have[3] $J_{c\alpha}^L(x) = \delta_{c\alpha}$ corresponding to a linear activation function at the final layer. The two components of the gradient then simplify to

$$
\begin{aligned}
\frac{\partial \mathcal{L}_{dep}}{\partial w_{\alpha\beta}^L} &= -\left\langle P_L(\alpha|x_i)\left(\phi_\beta^{L-1}(x_i) - \bar{\phi}_\beta^{L-1}\right) \right\rangle_\mathcal{D} \\
\frac{\partial \mathcal{L}_{ind}}{\partial w_{\alpha\beta}^L} &= +\left\langle P_M(\alpha|x_i)\left(\phi_\beta^{L-1}(x_i) - \bar{\phi}_\beta^{L-1}\right) \right\rangle_\mathcal{D}
\end{aligned}
\tag{16}
$$

With $\bar{\phi}_\beta^{L-1} \equiv \left\langle \phi_\beta^{L-1}(x_i) \right\rangle_\mathcal{D}$

### A.2.3 DERIVATION OF CLAIM 3

For small weight initialization, the network can be approximately linearized around zero weights[4]. Using the fact that the label independent part vanishes when the network outputs a uniform distribution as shown in section

---

[3]$\delta_{ij}$ is the Kronecker delta

[4]For ReLU networks, this is slightly more complex as the network is not differentiable with zero weights.

A.2.2, we see that this term in the gradient scales with the size of the weights as the $\mathcal{O}(1)$ term in the expansion about zero weights is zero.

$$\left\|\left.\frac{\partial\mathcal{L}_{ind}}{\partial w_{\alpha\beta}^L}\right|_{init}\right\|_2 \approx \mathcal{O}\left(|w|\right) \tag{17}$$

Here, the notation $\cdot|_{init}$ denotes that the gradient is evaluated with the initial weights. Hence, near initialization, the label independent part of the gradient is small regardless of if it is computed from unpermuted or permuted examples.

To gain insight into how the label dependent part behaves at initialization, we can write the gradient to $\mathcal{O}(|w|)$ in the size of the weights. In this case, the $\mathcal{O}(1)$ term is also zero as $g_{c\alpha\beta}^l(x) = const = \bar{g}_{c\alpha\beta}^l$ at zero weights for all $c$, and hence $\nabla_\theta\mathcal{L}_{dep} = 0$ for zero weights. The leading order term is then the $\mathcal{O}(|w|)$ term:

$$\frac{\partial N_c(x)}{\partial w_{\alpha\beta}^l} = J_{c\alpha}^l \sum_{\gamma=1}^{d} M_{\beta\gamma}^{l-1} x_\gamma + \mathcal{O}(|w|^2) \tag{18}$$

Here, the Jacobian $J_{c\alpha}^l$ is no longer a function of $x$, $d$ is the dimensionality of the input data $x$, and the network up to layer $l-1$ has been linearized to $M_{\beta\gamma}^{l-1} x_\gamma$. The label dependent part of the gradient then has a simpler form

$$\frac{\partial\mathcal{L}_{dep}}{\partial w_{\alpha\beta}^L} \approx -\left\langle \sum_{c=1}^{P} P_L(c|x_i)\left(J_{c\alpha}^l \sum_{\gamma=1}^{d} M_{\beta\gamma}^{l-1} x_\gamma - \bar{g}_{c\alpha\beta}^l\right)\right\rangle_{\mathcal{D}} \tag{19}$$

where now, $\bar{g}_{c\alpha\beta}^l = \frac{1}{P}\sum_{c=1}^{P}\sum_{\gamma=1}^{d} J_{c\alpha}^l M_{\beta\gamma}^{l-1} \langle x_\gamma\rangle_{\mathcal{D}}$. By linearity of the expectations, this is

$$\frac{\partial\mathcal{L}_{dep}}{\partial w_{\alpha\beta}^L} \approx -\sum_{c=1}^{P}\sum_{\gamma=1}^{d}\left(J_{c\alpha}^l M_{\beta\gamma}^{l-1} \langle P_L(c|x_i)x_\gamma\rangle_{\mathcal{D}} - \bar{g}_{c\alpha\beta}^l\right) \tag{20}$$

Without loss of generality, for simplicity we will assume a centered dataset, which has $\langle x_\gamma\rangle_{\mathcal{D}} = 0$. This does not affect the following arguments, but allows for simpler notation. With this assumption, we have

$$\frac{\partial\mathcal{L}_{dep}}{\partial w_{\alpha\beta}^L} \approx -\sum_{c=1}^{P}\sum_{\gamma=1}^{d} J_{c\alpha}^l M_{\beta\gamma}^{l-1} \langle P_L(c|x_i)x_\gamma\rangle_{\mathcal{D}} \tag{21}$$

For many datasets, this is nonzero, as the data $x$ has some correlation with the labels, and so $\left|\langle P_L(c|x_i)x_\gamma\rangle_{\mathcal{D}}\right| > 0$. For datasets consisting of a mix of unpermuted and permuted examples, the cross correlation can be split into contributions from each part:

$$\langle P_L(c|x_i)x_\gamma\rangle_{\mathcal{D}} = \frac{|\mathcal{D}_u|}{|\mathcal{D}|}\langle P_L(c|x_i)x_\gamma\rangle_{\mathcal{D}_u} + \frac{|\mathcal{D}_p|}{|\mathcal{D}|}\langle P_L(c|x_i)x_\gamma\rangle_{\mathcal{D}_p} \tag{22}$$

Here, $|\mathcal{D}|$ is the size of the dataset, $\mathcal{D}_u$, $\mathcal{D}_p$ denotes the set of unpermuted, permuted examples, and $|\mathcal{D}_u|, |\mathcal{D}_p|$ denotes the number of unpermuted, permuted examples. Note we have $|\mathcal{D}_u| + |\mathcal{D}_p| = |\mathcal{D}|$.

For permuted examples, the label is assigned uniformly randomly and independent of the data. Assuming the labels are one-hot as in our experiment, the cross correlation $\langle P_L(c|x_i)x_\gamma\rangle_{\mathcal{D}_p}$ is simply the average of $x_\gamma$ over examples given a specific random label. By the central limit theorem, we have $|\langle P_L(c|x_i)x_\gamma\rangle_{\mathcal{D}_p}| \approx \mathcal{O}\left(\sqrt{\frac{P}{|\mathcal{D}_p|}}\right)$ for $P$ classes. With a fraction $\epsilon$ of permuted examples, we have $|\mathcal{D}_p| = \frac{\epsilon|\mathcal{D}|}{P}$. For the label dependent part of the gradient, this gives

$$\frac{\partial\mathcal{L}_{dep}}{\partial w_{\alpha\beta}^L} \approx -\frac{1}{P}\sum_{c=1}^{P}\sum_{\gamma=1}^{d} J_{c\alpha}^l M_{\beta\gamma}^{l-1}\left((1-\epsilon)\langle P_L(c|x_i)x_\gamma\rangle_{\mathcal{D}_u} + \mathcal{O}\left(\sqrt{\frac{\epsilon P}{|\mathcal{D}|}}\right)\right) \tag{23}$$

As the size of the dataset grows, $|\mathcal{D}| \to \infty$, the contribution from permuted examples vanishes.

Further splitting the label dependent gradient into contributions from unpermuted and permuted examples

$$\frac{\partial\mathcal{L}_{dep}}{\partial w_{\alpha\beta}^L} = \frac{\partial\mathcal{L}_{dep}^{unperm}}{\partial w_{\alpha\beta}^L} + \frac{\partial\mathcal{L}_{dep}^{perm}}{\partial w_{\alpha\beta}^L} \tag{24}$$

Provided $\epsilon < 1$, we obtain an ordering relationship for the relative size of label dependent part of the gradient computed on unpermuted and permuted examples for large datasets:

$$\left\|\left.\frac{\partial\mathcal{L}_{dep}^{unperm}}{\partial w_{\alpha\beta}^L}\right|_{init}\right\|_2 \geq \left\|\left.\frac{\partial\mathcal{L}_{dep}^{perm}}{\partial w_{\alpha\beta}^L}\right|_{init}\right\|_2 \approx \mathcal{O}\left(\sqrt{\frac{1}{|\mathcal{D}|}}\right) \tag{25}$$

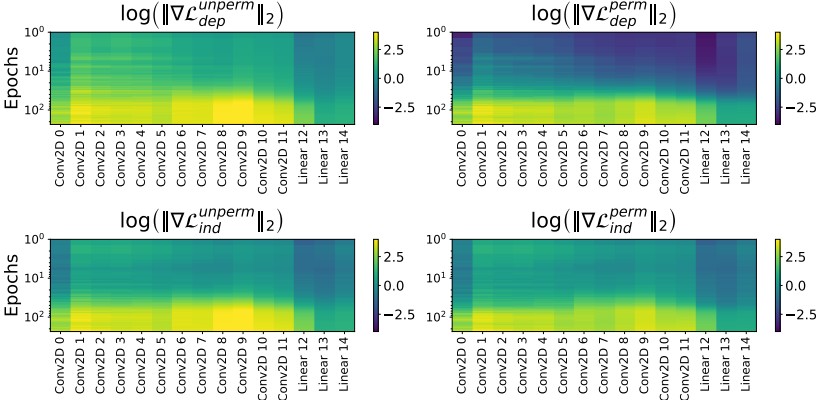

Figure A.1: **Top**: $\ell^2$ norm of the label dependent gradient calculated for unpermuted examples ($\|\nabla\mathcal{L}_{dep}^{unperm}\|_2$) and permuted examples ($\|\nabla\mathcal{L}_{dep}^{perm}\|_2$). Note that the label independent gradient has a very small norm in the early epochs for permuted examples. **Bottom**: $\ell^2$ norm of the label independent gradient calculated for unpermuted examples ($\|\nabla\mathcal{L}_{ind}^{unperm}\|_2$) and permuted examples ($\|\nabla\mathcal{L}_{ind}^{perm}\|_2$). The trends are overall similar between permuted and unpermuted examples, though somewhat higher in magnitude for unpermuted examples in the later epochs.

Where the degree by which the left hand side is greater than the right hand side is determined by the cross covariance matrix $\langle P_L(c|x_i)x_\gamma\rangle_{\mathcal{D}}$.

In summary, we see in Eq. 17 that the label independent part of the gradient is small early in training given small weight initialization. This part of the gradient behaves similarly for permuted and unpermuted examples. In Eq. 25 we see that the contribution to the label dependent part of the gradient from permuted examples vanishes for large datasets, while the contribution from unpermuted examples does not provided the cross correlation between input features and labels is nonzero. This suggests that with small weight initialization, the gradient descent dynamics initially ignores the labels of permuted examples.

Figure A.1 shows a breakdown of how the two components of the gradient computed on both unpermuted and permuted examples evolve over the course of training for the different layers of the VGG16 model trained on CIFAR-100. We see that the label dependent part behaves qualitatively differently for the unpermuted examples than for the permuted examples, as the permuted examples give close to zero contribution early in training in agreement with Eq. 25. The label independent part of the gradient shows similar trends between unpermuted and permuted examples, though in the final epochs, the unpermuted examples have a slightly larger label independent gradient indicating slightly greater model confidence on these examples. As the label dependent and label independent parts of the gradient have differing signs, they compete with each other and cancel when the loss is minimized, but are not independently zero and in fact grow during training. The slightly larger label independent gradient for unpermuted examples is balanced by a corresponding slightly larger label dependent gradient at the end of training.

## A.3 A SHORT COMMENT ON THE RANDOMIZATION OF LABELS

In our experiments, we define the permuted examples as the set of all examples which had their labels randomly permuted. Due to random chance, some fraction of the label permutations will result in the randomly assigned label being identical to the training (or 'correct') label. This choice winds up being important, as the inclusion of this possibility implies that on average, the input data contains no information about the correct label, so they must be memorized during training. If the set of permuted examples were taken to be the set of examples that have labels differing from their training labels, this would no longer be true. For example, given a dataset with classes $a, b, c$, if seeing the permuted label $a$ for a datapoint $x_i$ implies that $x_i$ definitely does not actually belong to class $a$, features useful for predicting the permuted label $a$ from $x_i$ may also be useful for predicting that $x_i$ belongs to $b$ or $c$ and vice versa. In this case, memorization is not the only way to solve the problem.

## A.4 UMAP VISUALIZATION FOR EARLIER LAYERS

In the main text, we show an example of a UMAP visualization for the final layer features in VGG16 trained on the CIFAR-100 dataset. This type of visualization is also possible in the early layers, although the lower separability results in a noisier visualization. Nevertheless, as shown in Fig. SI A.2 for layer 15 of VGG16, some

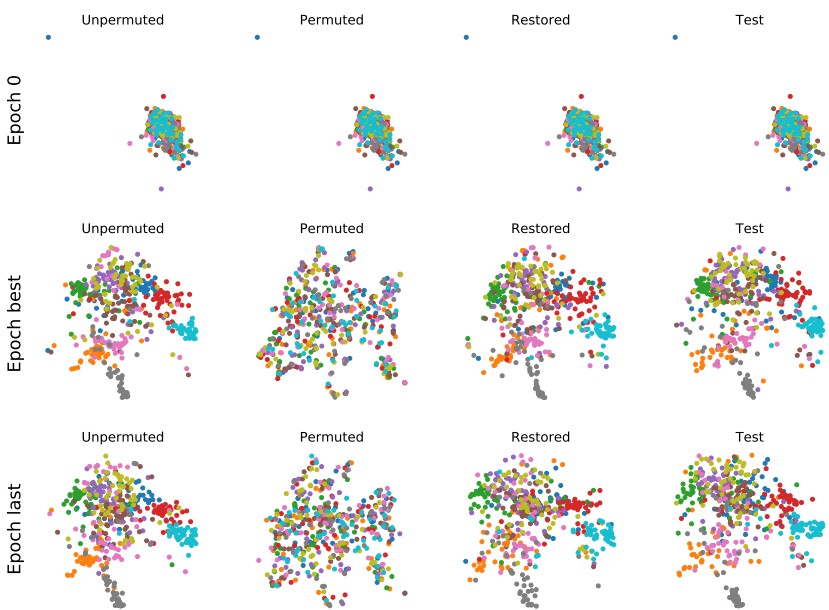

Figure A.2: UMAP visualization of the layer 15 features in VGG16 with $f = 50\%$ at initialization, best epoch, and final epoch. At the best and last epoch, unpermuted, restored, and test manifolds show similar structure, implying that the network has learned the features shared between the two datasets.

information can be gleaned using this method. We see that in this intermediate layer, unpermuted, restored, and test examples appear to behave similarly even after the final epoch of training. The memorization observed for the permuted examples in the final layers of the network does not appear here, and instead the restored examples show some separability. This is consistent with the quantitative trends shown in the MGMs in the main text, where we found that the early and intermediate layers do not show evidence of memorization and instead learn features that group examples according to their true label.

## A.5 EXPERIMENT WITH SYNTHETIC DATA

As an early experiment, we created a synthetic manifold dataset to test our understanding and also examine non-CNN architectures such as multi-layer perceptrons. This dataset consists of spherical manifolds, allowing a finer degree of control over the manifold geometry and initial separability of the data. We constrained the centers of the spheres to be diametrically opposed pairs which are orthogonal to all other pairs, ensuring that this data is linearly separable if the sphere radius is less than $r = 1$. Otherwise, the spheres overlap and linear separation is impossible. For the results presented in the main text, we used values of $r = 5$, $d = 30$, and $D = 1024$, well into the linearly inseparable regime to better mimic complex datasets.

An example of a pair of classes used in the experiments shown in the main text can be seen in Fig. A.3 (Left). Note that the constraint that the sphere centers to be diametrically opposed pairs implies that only the shared direction parallel to the centers contains linearly decodable information about the class ID, as the data from each class is mean zero along all other directions. Figure A.3 demonstrates this for classes $0, 1$, showing partial separability along feature dimension $0$, and near i.i.d. data for $4$ other features. The remaining features are similar, and other class pairs have a similar structure.

We trained a 5 layer feed-forward network on this dataset with 1024-dimensional linear layers interspersed with ReLU nonlinearities. This network has $\approx 5.2 \times 10^6$ parameters, over two orders of magnitude larger than the number of training points, such that complete memorization is easy. We see similar trends to those shown in the main text in train and test accuracy, as well as the accuracy on unpermuted, permuted and restored examples as shown in Fig. A.3 (Right).

Figure A.4 (Top) shows the best epoch results for the MGMs on the feedforward model trained on synthetic data. The trends here are similar, in that we see evidence for increasing separability of unpermuted, restored, and test manifolds across the layers of the network, but very little evidence of untangling of permuted manifolds at this point. Again, early layer features for unpermuted, restored, and test manifolds are all similar, with differences

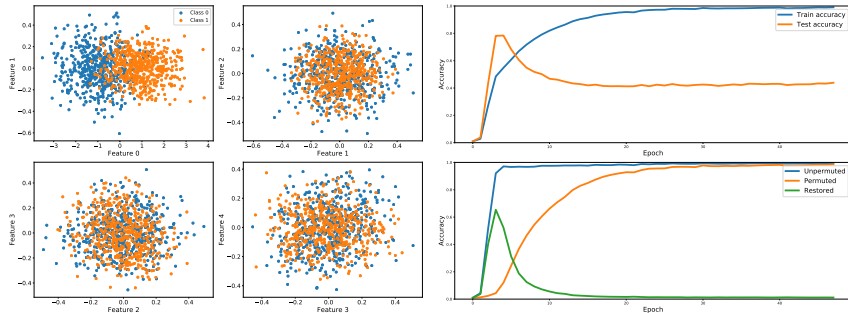

Figure A.3: **Left**: Classes 0 and 1 from the synthetic dataset. Note only a subset of features are useful for separation, the rest of the features contain no information about class 0 vs. 1. **Right**: Accuracy trends for a five layer feedforward model trained on this dataset with $\epsilon = 0.5$.

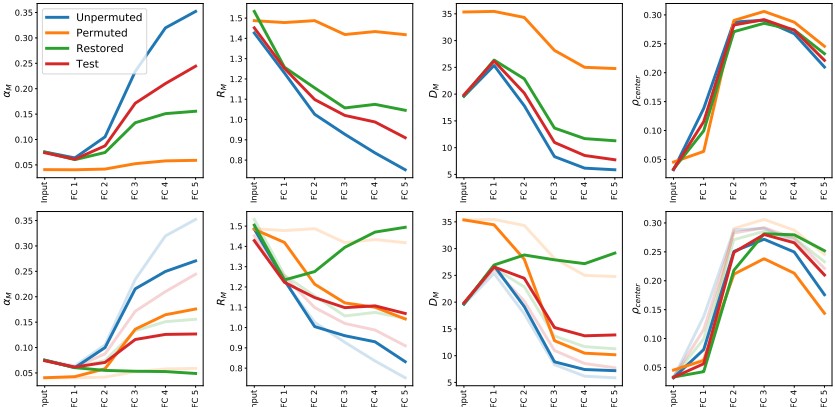

Figure A.4: **A**: Manifold analysis on the feedforward model trained with synthetic data at best epoch. **B**: Manifold analysis on the feedforward model trained with synthetic data at the last epoch.

becoming more pronounced in the later layers. Note that due to how the centers are initialized for the data, the center correlation $\rho_{center}$ is near zero at the input.

Figure A.4 (Bottom) shows the results for the MGMs on the feedforward model trained on synthetic data for the final epoch of training. In contrast to Fig. A.4, we now see rising separability of permuted manifolds after layer 2, and restored manifolds initially become more separable, before the effect saturates in the final three layers. Test manifolds now show very little separability, indicating the memorization of the permuted examples has significantly impacted the generalization performance of the network.

## A.6   MFTMA RESULTS FOR ADDITIONAL MODELS AND DATASETS AT $\epsilon = 0.5$

Here, we include the results of the experiments described in the main text on the remaining models evaluated.

Figure A.5(A) shows the results for the MGMs on AlexNet trained on CIFAR-100 at the best epoch of training. Again, unpermuted, restored, and test manifolds behave nearly identically and become increasingly separable deeper into the network, with the exception of the final two layers 7 and 8. This decrease in the final layers is perhaps part of the reason why this model performs worse than VGG16 and ResNet18 on CIFAR-100, and is partially explained by the large label dependent part of the gradient on permuted examples shown in Fig. A.12 in the final two layers, which indicates little preference for learning correctly labeled examples in these last two layers.

Figure A.5(B) shows the results for the MGMs on AlexNet trained on CIFAR-100 at the final epoch of training. Now, permuted manifolds begin to become separable in the final three layers of the model, 6, 8, and 7. Though, perhaps owing to the smaller model capacity of AlexNet compared to VGG16, this effect is smaller than we observed in other models. This is also reflected in the smaller difference between unpermuted and restored manifolds in the final layer. We note that this model also took many more epochs to memorize the data than VGG16 or ResNet18 as shown in the training curves in the main text.

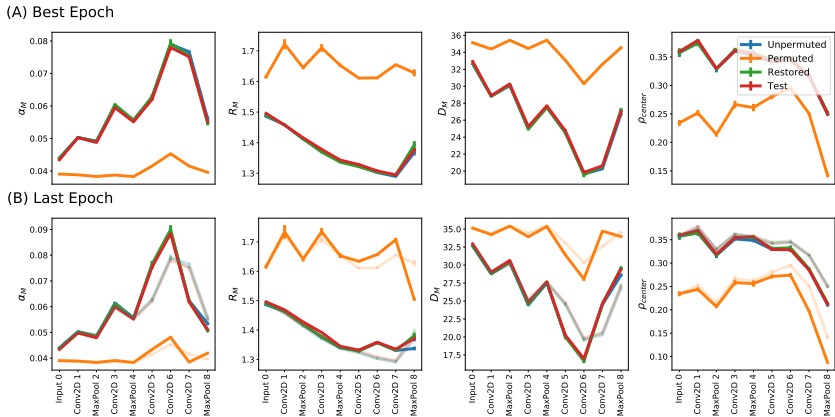

Figure A.5: **A**: Manifold analysis on AlexNet trained on CIFAR-100 at best epoch. **B**: Manifold analysis on AlexNet trained on CIFAR-100 at the last epoch. Error bars indicate 95% confidence intervals computed over 10 training runs with different random seeds.

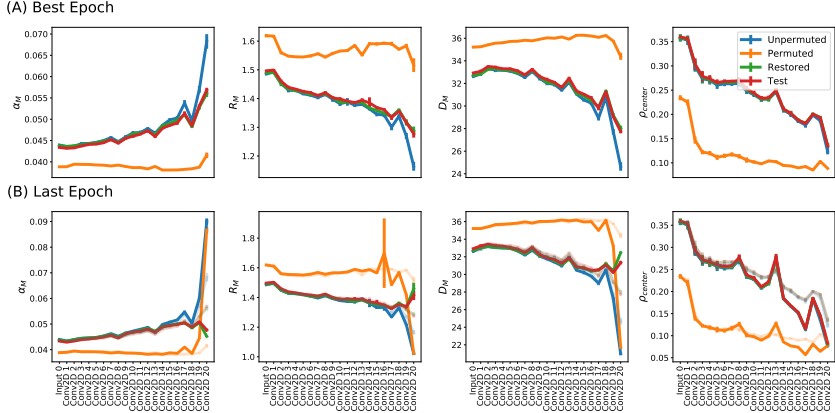

Figure A.6: **A**: Manifold analysis on ResNet18 trained on CIFAR-100 at best epoch. **B**: Manifold analysis on ResNet18 trained on CIFAR-100 at the last epoch. Error bars indicate 95% confidence intervals computed over 10 training runs with different random seeds.

Figure A.6(A) shows the results for the MGMs on ResNet18 trained on CIFAR 100. Here, we only show the metrics computed on the 2D Conv. + ReLU layer in the residual block and omit the analysis on the skip connections for fair comparison with other models. Again, we see that at the best epoch of training, unpermuted, restored and test manifolds all become increasingly separable deeper into the network, with restored and test manifolds behaving similarly. Permuted manifolds exhibit low separability at every layer, indicating that these examples have yet to be learned at any layer.

Figure A.6(B) shows the MGMs computed for ResNet18 trained on CIFAR-100 at the final epoch of training. Again, permuted manifolds show increasing separability in the final two layers (labeled 7 and 8 here) of the model, while the manifold capacity remains at the lower bound at all earlier layers. Unpermuted, restored, and test manifolds behave identically up to layer 7, numerically very similar to the best epoch behavior, indicating little memorization in the early layers. In the final two layers, unpermuted manifolds show a more favorable geometry than restored or test manifolds.

Figure A.7(A) shows the MGMs computed on ResNet18 trained on the Tiny ImageNet dataset at the best epoch of training. Interestingly, we note that test manifolds exhibit a more favorable geometry for classification than unpermuted, restored, or permuted manifolds in many layers of this model. This is likely due to the test set consisting of easier examples than the training set on average. Aside from this, the observed trends are similar to the ones observed in other models and datasets in that unpermuted, restored, and test manifolds show numerically similar trends, and increase in manifold capacity (and decrease in manifold radius and dimension) across the layers of the network, while the permuted manifolds remain at the manifold capacity lower bound.

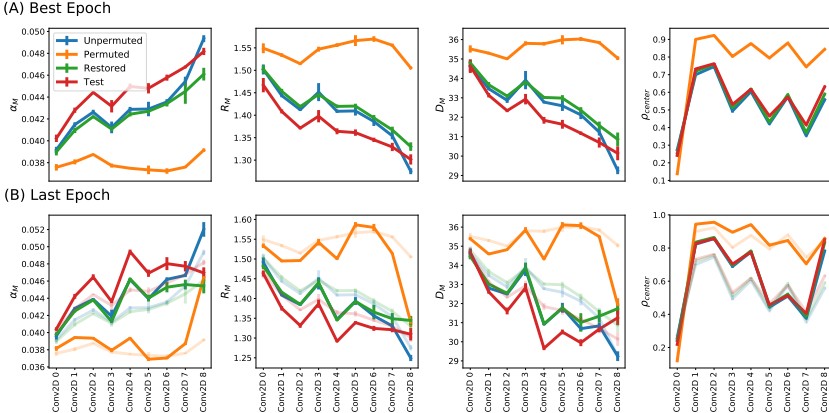

Figure A.7: **A**: Manifold analysis on ResNet18 trained on CIFAR-100 at best epoch. **B**: Manifold analysis on ResNet18 trained on CIFAR-100 at the last epoch. Error bars indicate 95% confidence intervals for the MGMs.

Finally, Fig. A.7(B) shows the MGMs computed on ResNet18 trained on the Tiny ImageNet dataset at the final epoch of training. Again we see that permuted manifolds show increased separability in the final two layers of the network (7 and 8) while restored and test manifolds show an initially increasing manifold capacity with layers, followed by a saturation and decrease in layers 7 and 8 coinciding with the rise in permuted manifold capacity. Up to layer 6, the trends for unpermuted, restored, and test manifolds are numerically very similar to those seen at the best epoch of training, indicating little memorization has occurred in these layers. This can also be seen from the permuted manifold capacity which remains near the lower bound in these layers.

## A.7    MANIFOLD CAPACITY FOR CIFAR-100 MODELS AT VARYING $\epsilon$

The trends presented in the main text hold for a wide range of permutation fractions $\epsilon$. Figure A.8 shows a summary of the manifold capacity results for the three models trained on CIFAR-100 for $\epsilon$ between 0 and 1. Figure A.8(A) shows the manifold capacity measured at the first layer for the three models. As discussed in the main text, in the early layers of the models we see no significant difference between unpermuted, restored, and test manifolds, while the separability of permuted manifolds remains near the lower bound. These trends are stable across permutation fractions, though we see a slight increase in manifold capacity with $\epsilon$, but this trend is within error bars for most models. Continuing to the middle layers shown in Figure A.8(B), there is still no difference between unpermuted, restored, and test manifolds, and the separability of permuted manifolds remains near the lower bound. We do see now a slight decrease in manifold capacity for larger values of $\epsilon$.

In the last layers, the situation is quite different as A.8(C) shows for the final layer in each model. Here, unpermuted manifolds show higher separability than restored or test, and for the two larger models, VGG16 and ResNet18, permuted manifolds show significant separability indicating memorization. We note also that the manifold capacity measured on test examples decreases significantly with increasing permutation fraction $\epsilon$ as the memorization increasingly degrades the model performance.

## A.8    ADDITIONAL LAYER REWINDING PLOTS

We also carried out the layer rewinding experiment on AlexNet and ResNet18 trained on CIFAR-100, and found similar trends as seen on VGG16 shown in the main text. Figures A.9 and A.10 show the results averaged over several training runs. For the case of ResNet18, we show every third convolutional layer, but include the final two layers to show the effect. We see that rewinding the last layer in the case of AlexNet corrects a significant fraction of the decrease in test accuracy, while in the case of the deeper ResNet18, the final two layers show an effect as in VGG16. In the case of ResNet18, the final layer obtains its peak value at epoch 0, a result corresponding to the behavior observed in (Zhang et al., 2019).

Finally, we include Fig. A.11 as an updated version of the plot in the main text for VGG16 including additional runs with different random seeds. We find the trends in the main text are robust across different seeds.

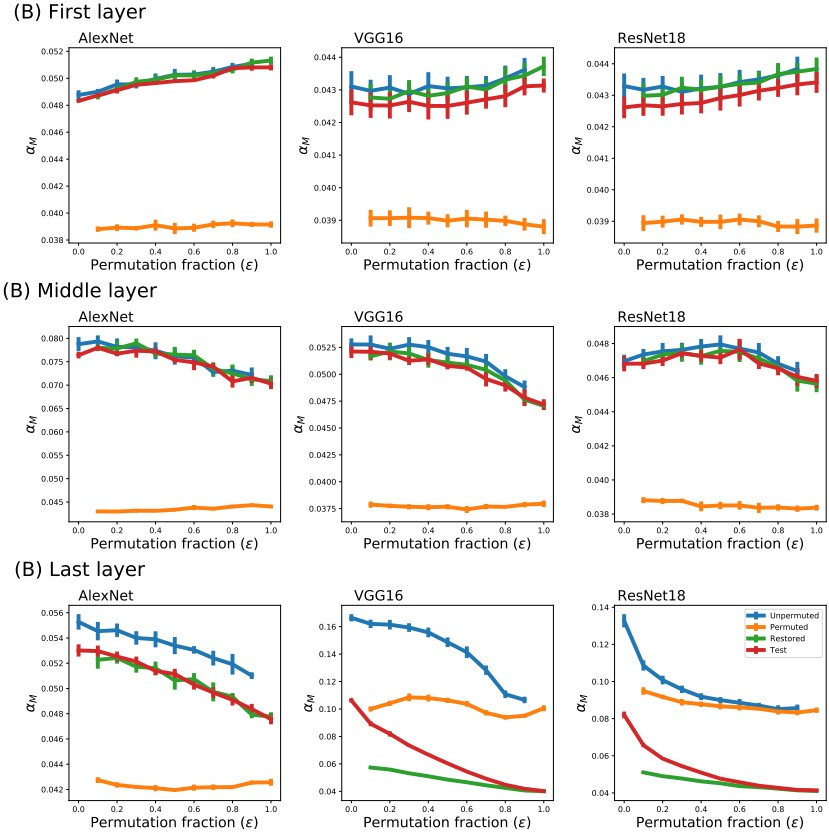

Figure A.8: **A** Manifold capacity for the first convolutional layer of AlexNet, VGG16, and ResNet18 computed for varying $\epsilon$. **B**: Manifold capacity for the middle convolutional layer. **C**: Manifold capacity for the final layer. Error bars indicate 95% confidence intervals computed over 10 training runs with different seeds.

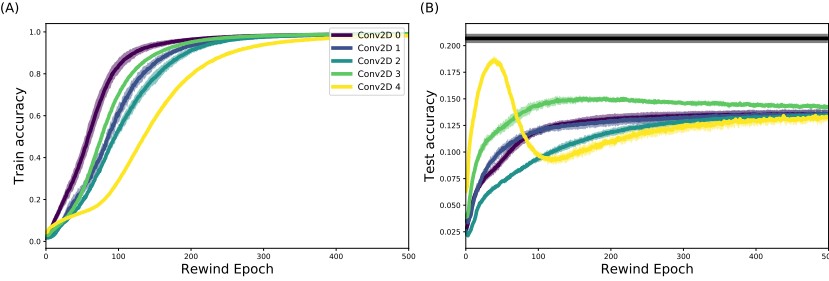

Figure A.9: **A**: Accuracy on the train set for AlexNet trained for 1000 epochs with an individual layer's parameters rolled back to an earlier epoch (rewind epoch). **B**: Corresponding results measured on the test set. Horizontal line indicates the peak generalization performance obtained by early stopping. Shaded region indicates the 95% confidence interval obtained over 10 training runs with different random seeds.

## A.9    ADDITIONAL GRADIENT PLOTS

Here, we show the results of the gradient analysis on AlexNet trained on CIFAR-100, ResNet18 trained on CIFAR-100, and ResNet18 trained on Tiny ImageNet. We find that these results are consistent with the results shown in the main text on VGG16 trained on CIFAR-100.

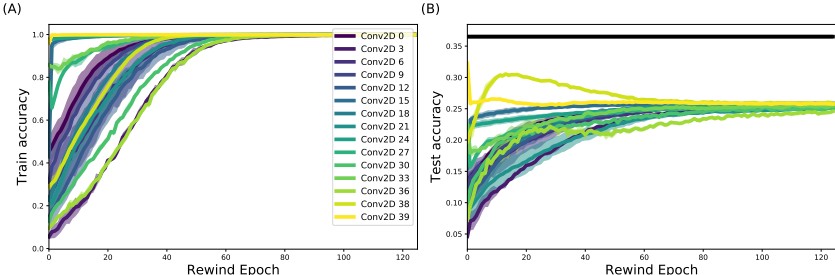

Figure A.10: **A**: Accuracy on the train set for ResNet18 trained for 1000 epochs with an individual layer's parameters rolled back to an earlier epoch (rewind epoch). **B**: Corresponding results measured on the test set. Horizontal line indicates the peak generalization performance obtained by early stopping. Shaded region indicates the 95% confidence interval obtained over 4 training runs with different random seeds.

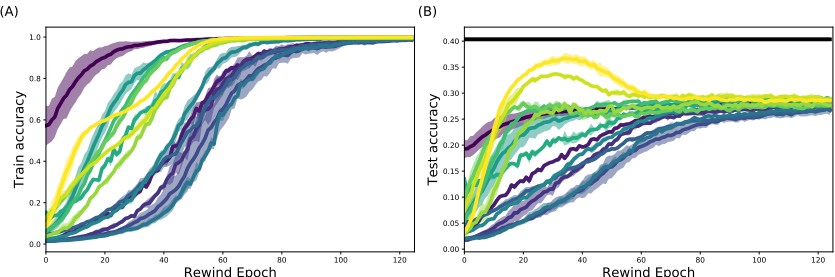

Figure A.11: **A**: Accuracy on the train set for VGG16 trained for 1000 epochs with an individual layer's parameters rolled back to an earlier epoch (rewind epoch). **B**: Corresponding results measured on the test set. Horizontal line indicates the peak generalization performance obtained by early stopping. Shaded region indicates the 95% confidence interval obtained over 3 training runs with different random seeds.

Figure A.12 shows the norm of the total gradient for AlexNet trained on CIFAR-100 computed on all examples as well as the unpermuted and permuted examples independently. No clear differences in sizes between these gradients is observed, and we do not see vanishing gradients in the early layers in the later epochs of training.

Figure A.12 shows the relative size (log ratios) of the label dependent and label independent components of the gradients on unpermuted and permuted examples for AlexNet trained on CIFAR-100. As discussed in the main text and in Sec. A.2, in the early epochs, the label dependent contribution from unpermuted examples dominates the label dependent part from the permuted examples. However, in this case we do note that in the final two layers (4 and 5 here), the permuted examples do contribute a large label dependent part, which is perhaps related to the somewhat decreased manifold capacity in this model shown in Fig. A.5 and Fig. A.5.

Figure A.13 shows the norm of the total gradient for ResNet18 trained on CIFAR-100 computed on all examples as well as the unpermuted and permuted examples independently. Again, we do not see evidence for vanishing gradients in the early layers or late epochs. We do note that the gradient on unpermuted examples is slightly larger than the gradient from permuted examples, although the difference is relatively small.

Figure A.13 shows the relative size (log ratios) of the label dependent and label independent components of the gradients on unpermuted and permuted examples for ResNet18 trained on CIFAR-100. The label dependent part of the gradient for unpermuted examples again dominates the label dependent part from permuted samples in the early epochs (up to epoch 150 after which they are of comparable size). This is true for all layers of the network.

Figure A.14 shows the norm of the total gradient for ResNet18 trained on Tiny ImageNet computed on all examples as well as the unpermuted and permuted examples independently. The gradient trends here are somewhat different than observed in the CIFAR-100 trained networks, although qualitatively similar. Specifically, we do not see evidence for vanishing gradients in the early layers or late epochs. Again, the gradient on unpermuted examples is slightly larger than the gradient from permuted examples, although the difference is relatively small. The largest difference from the CIFAR-100 trained networks is that the gradient in the first layer is much larger than the later layers.

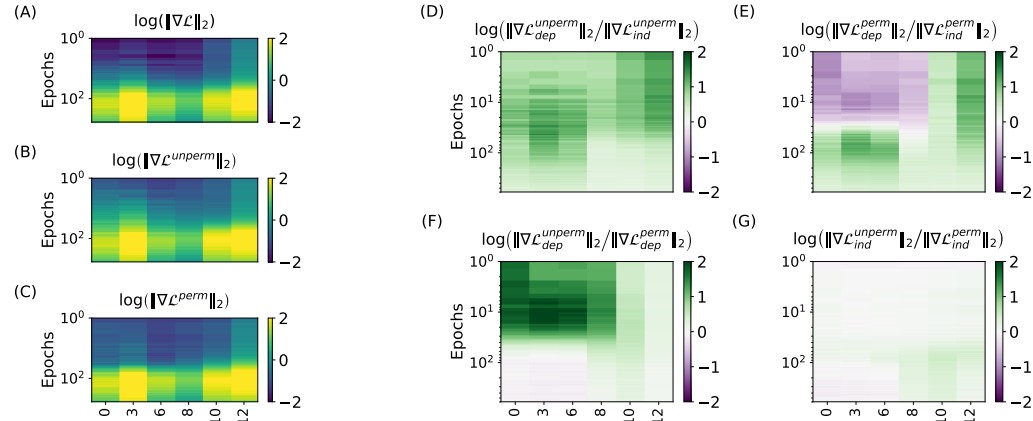

Figure A.12: **(A-C)**: $\ell^2$ norm of the gradient of the loss over the different layers and epochs of training for AlexNet, computed on (A) all samples, (B) unpermuted samples only, and (C) permuted samples only. **(D-G)**: Log ratio of gradients. **D**: For unpermuted samples, green indicates that the label dependent ($dep$) part is larger than the label independent part ($ind$). **E**: For permuted examples, purple indicates that $dep < ind$. **F**: For the dependent part, those computed on the unpermuted examples dominate that of permuted examples. **G**: For the independent part, the two components are roughly equal.

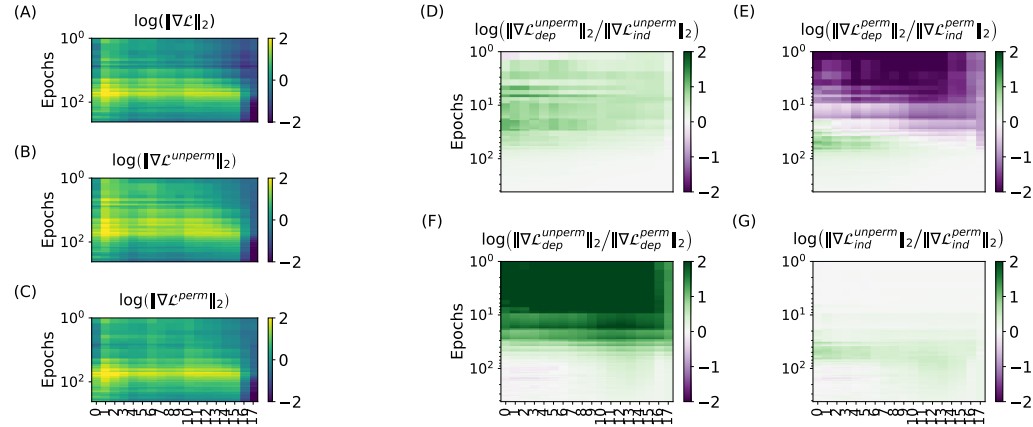

Figure A.13: **(A-C)**: $\ell^2$ norm of the gradient of the loss over the different layers and epochs of training for ResNet18 trained on CIFAR-100, computed on (A) all samples, (B) unpermuted samples only, and (C) permuted samples only. **(D-G)**: Log ratio of gradients. **D**: For unpermuted samples, green indicates that the label dependent ($dep$) part is larger than the label independent part ($ind$). **E**: For permuted examples, purple indicates that $dep < ind$. **F**: For the dependent part, those computed on the unpermuted examples dominate that of permuted examples. **G**: For the independent part, the two components are roughly equal.

Figure A.14 shows the relative size (log ratios) of the label dependent and label independent components of the gradients on unpermuted and permuted examples for ResNet18 trained on Tiny ImageNet. Again, the label dependent term from the unpermuted examples is much larger in magnitude than the label dependent term from the permuted examples in the early epochs. As in the case of AlexNet, this effect is smaller in the later layers, as the permuted examples produce a non-negligible label dependent gradient in these layers.

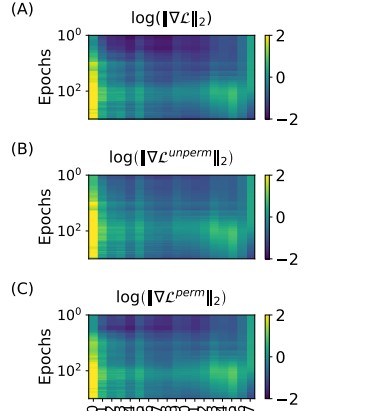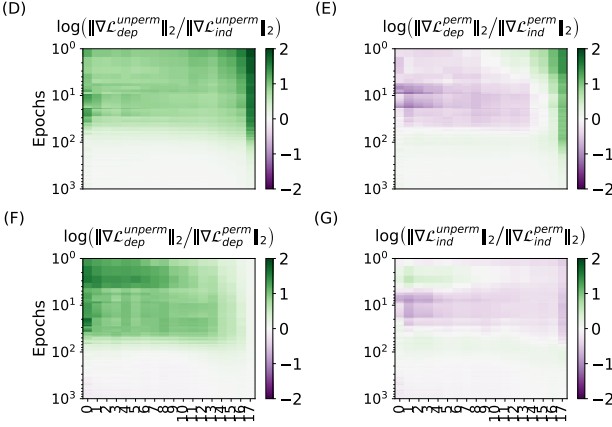

Figure A.14: **(A-C)**: $\ell^2$ norm of the gradient of the loss over the different layers and epochs of training for ResNet18 trained on Tiny ImageNet, computed on (A) all samples, (B) unpermuted samples only, and (C) permuted samples only. **(D-G)**: Log ratio of gradients. **D**: For unpermuted samples, green indicates that the label dependent ($dep$) part is larger than the label independent part ($ind$). **E**: For permuted examples, purple indicates that $dep < ind$. **F**: For the dependent part, those computed on the unpermuted examples dominate that of permuted examples. **G**: For the independent part, the two components are roughly equal.

## A.10 MODEL ARCHITECTURES AND TRAINING DETAILS

All networks were trained using the Adam optimizer with a learning rate of $1 \times 10^{-4}$ and a batch size of $1,024$. This choice of hyperparameters was made not to optimize for the generalization performance of any of the networks, but rather to provide a consistent training methodology across all network architectures and datasets. We note that at high fractions of label permutation (not shown in this work) typical training hyperparameters may not work well for training, and we found that this combination of decreased learning rate and increased batch size works across all fractions of label permutation at the cost of decreasing generalization performance somewhat. Here, we give a complete specification for the network architectures used. Note that in some cases (particularly for the ResNets where we omit analysis of skip connections) the layer numbers given here may differ from those in the previously shown results. Additionally, when evaluating the MGMs, we evaluate using the representations following a nonlinearity.

Table 1: FeedForward network

| Layer | Type | Size |
|---|---|---|
| 0 | Input | 1024 features |
| 1 | Linear + ReLU | $1024 \times 1024$ |
| 2 | Linear + ReLU | $1024 \times 1024$ |
| 3 | Linear + ReLU | $1024 \times 1024$ |
| 4 | Linear + ReLU | $1024 \times 1024$ |
| 5 | Linear + ReLU | $1024 \times 1024$ |
| 6 | Linear | $1024 \times 100$ |

Table 2: Modified AlexNet

| Layer | Type | Size |
|---|---|---|
| 0 | Input | $3 \times 32 \times 32$ |
| 1 | 2D Conv. + ReLU | $64\ 11 \times 11$ filters |
| 2 | 2D MaxPool | $2 \times 2$ stride 2 |
| 3 | 2D Conv. + ReLU | $192\ 5 \times 5$ filters |
| 4 | 2D MaxPool | $2 \times 2$ stride 2 |
| 5 | 2D Conv. + ReLU | $384\ 3 \times 3$ filters |
| 6 | 2D Conv. + ReLU | $256\ 3 \times 3$ filters |
| 7 | 2D Conv. + ReLU | $1024\ 3 \times 3$ filters |
| 8 | 2D MaxPool | $2 \times 2$ stride 2 |
| 9 | Linear | $1024 \times 100$ |

Table 3: VGG16 for CIFAR-100

| Layer | Type | Size |
|---|---|---|
| 0 | Input | $3 \times 32 \times 32$ |
| 1 | 2D Conv. + BatchNorm + ReLU | $64\ 3 \times 3$ filters |
| 2 | 2D Conv. + BatchNorm + ReLU | $64\ 3 \times 3$ filters |
| 3 | 2D MaxPool | $2 \times 2$ stride 2 |
| 4 | 2D Conv. + BatchNorm + ReLU | $128\ 3 \times 3$ filters |
| 5 | 2D Conv. + BatchNorm + ReLU | $128\ 3 \times 3$ filters |
| 6 | 2D MaxPool | $2 \times 2$ stride 2 |
| 7 | 2D Conv. + BatchNorm + ReLU | $256\ 3 \times 3$ filters |
| 8 | 2D Conv. + BatchNorm + ReLU | $256\ 3 \times 3$ filters |
| 9 | 2D Conv. + BatchNorm + ReLU | $256\ 3 \times 3$ filters |
| 10 | 2D MaxPool | $2 \times 2$ stride 2 |
| 11 | 2D Conv. + BatchNorm + ReLU | $512\ 3 \times 3$ filters |
| 12 | 2D Conv. + BatchNorm + ReLU | $512\ 3 \times 3$ filters |
| 13 | 2D Conv. + BatchNorm + ReLU | $512\ 3 \times 3$ filters |
| 14 | 2D MaxPool | $2 \times 2$ stride 2 |
| 15 | 2D Conv. + BatchNorm + ReLU | $512\ 3 \times 3$ filters |
| 16 | 2D Conv. + BatchNorm + ReLU | $512\ 3 \times 3$ filters |
| 17 | 2D Conv. + BatchNorm + ReLU | $512\ 3 \times 3$ filters |
| 18 | 2D MaxPool | $2 \times 2$ stride 2 |
| 19 | Linear + ReLU + Dropout | $512 \times 4096$ |
| 20 | Linear + ReLU + Dropout | $4096 \times 4096$ |
| 21 | Linear | $4096 \times 100$ |

Table 4: ResNet18 for CIFAR-100

| Layer | Type | Size |
|---|---|---|
| 0 | Input | $3 \times 32 \times 32$ |
| 1 | 2D Conv. + BatchNorm + ReLU | $64\ 3 \times 3$ filters |
| 2 | 2D Conv. + BatchNorm + ReLU | $64\ 3 \times 3$ filters |
| 3 | 2D Conv. + BatchNorm + ReLU | $64\ 3 \times 3$ filters |
| 4 | 2D Conv. + BatchNorm + ReLU | $64\ 3 \times 3$ filters |
| 5 | 2D Conv. + BatchNorm + ReLU | $64\ 3 \times 3$ filters |
| 6 | 2D Conv. + BatchNorm + ReLU | $128\ 3 \times 3$ filters |
| 7 | 2D Conv. + BatchNorm | $128\ 3 \times 3$ filters |
| 8 | 2D Conv. + BatchNorm + ReLU | $128\ 1 \times 1$ filters |
| 9 | 2D Conv. + BatchNorm + ReLU | $128\ 3 \times 3$ filters |
| 10 | 2D Conv. + BatchNorm + ReLU | $128\ 3 \times 3$ filters |
| 11 | 2D Conv. + BatchNorm + ReLU | $256\ 3 \times 3$ filters |
| 12 | 2D Conv. + BatchNorm | $256\ 3 \times 3$ filters |
| 13 | 2D Conv. + BatchNorm + ReLU | $256\ 1 \times 1$ filters |
| 14 | 2D Conv. + BatchNorm + ReLU | $256\ 3 \times 3$ filters |
| 15 | 2D Conv. + BatchNorm + ReLU | $256\ 3 \times 3$ filters |
| 16 | 2D Conv. + BatchNorm + ReLU | $512\ 3 \times 3$ filters |
| 17 | 2D Conv. + BatchNorm | $512\ 3 \times 3$ filters |
| 18 | 2D Conv. + BatchNorm + ReLU | $512\ 1 \times 1$ filters |
| 19 | 2D Conv. + BatchNorm + ReLU | $512\ 3 \times 3$ filters |
| 20 | 2D Conv. + BatchNorm + ReLU | $512\ 3 \times 3$ filters |
| 21 | 2D Adaptive Average Pool | $1 \times 1$ |
| 22 | Linear | $512 \times 100$ |

Table 5: ResNet18 for TinyImageNet

| Layer | Type | Size |
|---|---|---|
| 0 | Input | $3 \times 64 \times 64$ |
| 1 | 2D Conv. + BatchNorm + ReLU | $64\ 7 \times 7$ filters |
| 2 | 2D MaxPool | $3 \times 3$ stride 2 |
| 3 | 2D Conv. + BatchNorm + ReLU | $64\ 3 \times 3$ filters |
| 4 | 2D Conv. + BatchNorm + ReLU | $64\ 3 \times 3$ filters |
| 5 | 2D Conv. + BatchNorm + ReLU | $64\ 3 \times 3$ filters |
| 6 | 2D Conv. + BatchNorm + ReLU | $64\ 3 \times 3$ filters |
| 7 | 2D Conv. + BatchNorm + ReLU | $128\ 3 \times 3$ filters |
| 8 | 2D Conv. + BatchNorm | $128\ 3 \times 3$ filters |
| 9 | 2D Conv. + BatchNorm + ReLU | $128\ 1 \times 1$ filters |
| 10 | 2D Conv. + BatchNorm + ReLU | $128\ 3 \times 3$ filters |
| 11 | 2D Conv. + BatchNorm + ReLU | $128\ 3 \times 3$ filters |
| 12 | 2D Conv. + BatchNorm + ReLU | $256\ 3 \times 3$ filters |
| 13 | 2D Conv. + BatchNorm | $256\ 3 \times 3$ filters |
| 14 | 2D Conv. + BatchNorm + ReLU | $256\ 1 \times 1$ filters |
| 15 | 2D Conv. + BatchNorm + ReLU | $256\ 3 \times 3$ filters |
| 16 | 2D Conv. + BatchNorm + ReLU | $256\ 3 \times 3$ filters |
| 17 | 2D Conv. + BatchNorm + ReLU | $512\ 3 \times 3$ filters |
| 18 | 2D Conv. + BatchNorm | $512\ 3 \times 3$ filters |
| 19 | 2D Conv. + BatchNorm + ReLU | $512\ 1 \times 1$ filters |
| 20 | 2D Conv. + BatchNorm + ReLU | $512\ 3 \times 3$ filters |
| 21 | 2D Conv. + BatchNorm + ReLU | $512\ 3 \times 3$ filters |
| 22 | 2D Adaptive Average Pool | $1 \times 1$ |
| 23 | Linear | $512 \times 200$ |

