# OpenReview forum: "On the geometry of generalization and memorization in deep neural networks"
_ICLR.cc/2021/Conference — ICLR 2021 Poster_

### Official Review · AnonReviewer1 · 2020-10-26

**Rating:** 7
**Confidence:** 4

**Review:**

This paper analyses memorization in DNNs, from the lens of memorization = fitting random labels, and finds that it seems to happen in later layers. These results are obtained using the MFTMA framework, a manifold analysis tool, testing geometric properties of individual layers. The analysis also attempts to explain why such a phenomenon exists, and makes a few interesting observations.
This paper does not propose any new algorithm, but instead settles some important questions by infirming or affirming past speculation on layer behaviour found in the literature.

I find three particularly interesting results in this paper:
- later layers seem to be responsible for memorization, while early layers seem to converge last but consistently learn "generalizing" features (although this may not be true for other architectures)
- increasing the dimensionality of the network to induce double descent _decreases_ the manifold dimensionality of the last layer. This is consistent with overparameterization making everything smoother/flatter and more easily disentangleable in the last layer.
- for examples with the wrong class, gradients initially vanish (due to destructive interference), which seems to be a driving force for the initial good generalization performance.

Downsides of the paper:
- The setting explored here is somewhat artificial, (1) the requirement on a high enough epsilon (random label proportion) may not represent real use of DNNs (I write this having seen Fig A.8; this is also a common criticism of double-descent results) (2) the models trained here don't seem to exceed 40% testing accuracy, again not necessarily representing real use of DNNs (this is a bit surprising considering even models from back in 2013 had above 60% accuracy on CIFAR100).
- Although the results of the paper do not hinge entirely on it, the reliance on MFTMA limits the interpretation somewhat: while an interesting tool, it's not clear to me that it allows us to make strong statements about the geometry of neural networks. In particular for the early layers, MFTMA may not be able to capture the geometry of features which might still be somewhat entangled yet possess a lot of richness.
- I have some issues with the presentation of the paper
- This paper does not really introduce a novel lens on generalization or significantly new ideas (although I'd argue it formalizes existing ideas and properly tests them empirically).

On the value of the contribution:
- I think having empirical evidence of the studied phenomena is valuable, more so than previous speculation on them.
- The empirical results presented here do open the door for new questions to be answered and may help focus the ongoing investigation of memorization and generalization in DNNs


Additional comments:
- Something seems wrong with Figure 2B-middle two columns. Aren't permuted and restored examples the same inputs X but with the corresponding Y changed? If this is the case, then their UMAP should be the same, the only difference between the second column and the third column should be the coloring of the points. I presume that the figure shows a different minibatch of Xs for these two columns; I would highly recommend not doing so and using the exact same inputs. It would be consistent with the text, and the presentation, e.g. Fig 1A.
- All Figures: the label fonts should be bigger. From the ICLR formatting guidelines: "use 10 point type [for text]", and "all artwork must be neat, clean, and legible." Having to zoom in and out to be able to read figures properly hurts accessibility and legibility, which detracts from the quality of the paper. Packing text, results, and figures in an 8-page document can be hard, but synthesizing information, including visual information contained in figures, is an essential skill in conveying knowledge.
-- Here are a few suggestions for this particular paper: Figure 1A seems unnecessary, the text conveys these 3 concepts clearly; Figure 1B is important and should take the entire width of the page, with legible fonts; Figure 2A's subplots all share the same X and Y axis, making their naming redundant and taking up space; Figure 2B's column labels are also repeated needlessly, taking up vertical space; Figure 3's X axis doesn't need individual layer name labels, and could be replaced with a single "Layer depth" label -- 3A and 3B also share this axis, leading to wasted vertical space (space that could be used to make fonts larger); idem for Figure 4A, individual layers do not need to be named, but rather the concept of layer depth can be conveyed with a properly labelled colorbar gradient -- 4CDE could be less wide and leave more horizontal space to make fonts larger.
- In Figure 5A, it's not immediately clear that the X axis are individual layers, the log(nabla) label should be on the colorbar rather than on top of the figure. I'd also suggest flipping the X and Y axis, as the X axis is typically used for time; this would allow there to have the three subplots side by side with a shared labelled colorbar on the right (matplotlib seems to be used here, see matplotlib.pyplot.subplots's sharex/sharey arguments for examples).

---

> ### Author Response · Authors · 2020-11-21
> **Response to Reviewer 1**
>
> We thank the reviewer for their thorough comments and suggestions to improve our manuscript, particularly for their detailed suggestions to improve the quality and readability of our figures. We also appreciate that the reviewer found several of the results interesting, and agrees that empirical study of these phenomena is valuable.
>
> In response to some of the reviewer’s concerns:
>
> -**"The setting explored here is somewhat artificial, (1) the requirement on a high enough epsilon (random label proportion) may not represent real use of DNNs (I write this having seen Fig A.8; this is also a common criticism of double-descent results) (2) the models trained here don't seem to exceed 40% testing accuracy, again not necessarily representing real use of DNNs (this is a bit surprising considering even models from back in 2013 had above 60% accuracy on CIFAR100)."**
>
> The reviewer is correct that the addition of label permutation noise and some training hyperparameters in our setup are different from standard training practice. These differences were necessary as we aim to study networks in which the memorization is unambiguous. Our experiment requires a set of randomly labeled examples which must be memorized if they are to be learned, and as noted in https://arxiv.org/abs/1705.10694, this requires slight modifications to the training procedure in the form of a larger batch size and a smaller learning rate. Both of these changes result in a decrease in generalization performance, though we note that the 40% test accuracy figure the review mentioned is the result of training on a dataset with 50% of it’s labels randomized. With smaller amounts of label noise we see much higher test accuracies, and we will include a plot of how test accuracy depends on the amount of label noise in the supplemental.
>
> -**"Although the results of the paper do not hinge entirely on it, the reliance on MFTMA limits the interpretation somewhat: while an interesting tool, it's not clear to me that it allows us to make strong statements about the geometry of neural networks. In particular for the early layers, MFTMA may not be able to capture the geometry of features which might still be somewhat entangled yet possess a lot of richness."**
>
> It is true that the MFTMA technique is more complex than other techniques for the analysis of representations (Linear probes, or comparative measures such as RSA, CCA, etc.) we believe it is the best tool for this purpose for two reasons:
> It is currently the only theory grounded technique that links geometric properties of the data (manifold radius, dimension) to a quantity relevant to classification performance (manifold capacity)
> Since the results obtained via MFTMA depends on the geometry of the manifold, it doesn’t require a second train/test split unlike linear probes, and converges quickly with the number of samples.
>
> We note that other measures such as intrinsic dimensionality [https://arxiv.org/pdf/1905.12784.pdf] or geodesics [https://arxiv.org/pdf/1511.06394.pdf] have been used to probe early layers' entanglement in the past, and we will add a discussion around this point in the manuscript in the next version. Nonlinear readout performance (such as quadratic readout, or with readouts with hidden layers) may also be an interesting measure of probing nonlinearly decodable information, in the highly entangled data regime. They are beyond the scope of the present work but certainly a very promising future direction.
>
> -**"Something seems wrong with Figure 2B-middle two columns. …"**
>
> We appreciate the reviewer for inspecting this plot so closely. The plot is correct even though the points plotted for permuted and restored examples are different. The difference here is due to the downsampling of the number of classes. Here we show only the first 10 classes of CIFAR 100 to avoid visual clutter, and so the set of permuted examples with labels between 1-10 is different from the set of restored examples with labels between 1-10. If one were to create this plot with all 100 classes then, as the reviewer correctly points out, that umap plots for permuted and restored examples would be the same with only the colors differing.
>
> -**Many suggestions to improve the presentation of our figures**
>
> We reiterate our appreciation for the well thought out list of suggestions for improving our figures. We have begun incorporating some of the suggested changes (see the revised manuscript, Figures 1, 2, 3, and 5) and will continue to work on the presentation of our figures to improve readability and visual clarity.

---

> > ### Comment · AnonReviewer1 · 2020-11-23
> > **Precisions**
> >
> > Thank you for your response.
> >
> > It's been suggested to me that a discussion of Coherent Gradient (https://openreview.net/forum?id=ryeFY0EFwS) may be relevant in this paper, due to the similarity of the intuitions that are gained from this paper and yours (although I'd argue _how_ those results are obtained is quite different).
> >
> > Figure 2B makes sense now, but, if I may suggest another improvement: to choose points to plot, you could choose the _intersection_ of points with permuted labels 1-10 and points with restored labels 1-10 (if that intersection is big enough) This should make the umap positions identical, avoid visual clutter, and still be a random selection due to the random process of permuting labels.

---

### Official Review · AnonReviewer3 · 2020-10-28
**Make interesting observations**

**Rating:** 7
**Confidence:** 3

**Review:**

###
Summary:
This paper investigates memorization in deep neural networks (DNNs).  Authors leverage mean field theoretic geometric analysis method (MFTMA) to analyze when and where memorization occurs in a DNN. Through empirical analysis, they show that i) generalizing feature are learned initially and that memorization happen later in training mostly in the top layers ii) we can mitigate memorization by rewinding the top layers parameters to earlier values. They also show that their MFTMA metrics can highlight the phenomena of double decent. Finally, they demonstrate that gradient descent initially ignores noisy example and focus on correctly labeled examples.

###
Reasons for score:
I lean toward acceptance. This paper makes interesting observation regarding memorization of deep network, it performs a good empirical study which provide enough evidences for the different claims.  Although, MFTMA could be a better explained in the main paper.

###
 Pros:
- As stated above, the paper makes interesting observation regarding memorization of deep network.
-  It performs a thorough empirical study.

###
Cons:
- I found it hard to understand MFTMA without referring to the appendix A. It would be nice to expand the explanation of MFTMA in the main paper. In addition, it would be good to further explain Fig 1. B which contains a lot of information.
- Does the observation scale to larger dataset such as ImageNet ?
- Experiments are run for only one seed.

---

> ### Author Response · Authors · 2020-11-21
> **Response to Reviewer 3**
>
>
> We are pleased that the reviewer found our observations interesting, and thank the reviewer for their suggestions for improving the presentation of the MFTMA method.
>
> In response to some of the reviewer’s comments:
>
> -**"I found it hard to understand MFTMA without referring to the appendix A. It would be nice to expand the explanation of MFTMA in the main paper. In addition, it would be good to further explain Fig 1. B which contains a lot of information."**
>
> We agree with the reviewers that the explanation of the MFTMA method in the main text should be expanded. In the posted revision, we used some of the additional page to further explain this technique and the intuition behind it. We also expanded the size of Fig. 1.B for better readability, and in the next iteration, we plan to expand the explanation about Figure 1B to give intuition behind the manifold radius, dimension, and capacity quantities and how they are computed.
>
> -**"Does the observation scale to larger dataset such as ImageNet?"**
>
> To demonstrate this, we also include results on the Tiny ImageNet which is larger than CIFAR100 both in the amount of data and in the size of the images. We also note that the behaviors we see when training without label noise on these datasets are similar to the trends observed when training on ImageNet without label noise, which can be seen in https://www.nature.com/articles/s41467-020-14578-5.
>
> -**"Experiments are run for only one seed."**
>
> We ran all our experiments with 10 random seeds, which varied the network initialization, the selection and labeling of the permuted examples, and the initialization of the MFTMA calculations. In most cases we found that our results were not very sensitive to this seed, hence error bars on our plots are fairly small and easy to miss. We will clarify this bit of methodology in the main text.

---

### Official Review · AnonReviewer2 · 2020-10-28
**New results providing an insight to understanding of generalization and memorization by DNNs**

**Rating:** 7
**Confidence:** 4

**Review:**

The authors apply MFTMA to DNNs trained on CIFAR with label noise to analyze their behaviors between generalization and memorization. Based on experimental results, they claim that what is involved in memorization are not lower layers but higher layers. This claim is convincing. Another claim that this is not caused by a vanishing gradient effect is plausible, too. I'm sure these results give some insights into understanding generalization and memorization by DNNs.

Questions.
Why do the authors consider only convolutional layers, not fully-connected layers, for the analyses? In the experiment of rewinding individual layers, the three FC layers are left untouched. Why?

Is MFTMA the only method that can examine/verify the above finding?

Comments.
At the first reading, I didn't understand what "restored examples" means, and it took me a while to understand it. The caption for Fig. A.7 has an error; CIFAR100 should be Tiny ImageNet.

---

> ### Author Response · Authors · 2020-11-21
> **Response to Reviewer 2**
>
> We are glad to hear that the reviewer found the claim regarding memorization in the later layers convincing, and also found it plausible that this is not due to vanishing gradients. We also wish to thank the reviewer for giving our manuscript such careful attention and finding a mislabeled figure in the supplemental information which we have corrected.
>
> Below, we respond to some of the questions the reviewer had:
>
> -**"Why do the authors consider only convolutional layers, not fully-connected layers, for the analyses? In the experiment of rewinding individual layers, the three FC layers are left untouched. Why?"**
>
> As is typical for image classification networks, most of the architectures we analyzed (AlexNet, and both of the ResNets) are constructed from purely convolutional, normalization, and pooling layers with a single FC layer at the output, while only VGG16 has a series of 3 FC layers before the output. For consistency across all networks we show results after convolutional layers or pooling layers, both of which can change the capacity while normalization layers can’t.
>
> For the experiment of rewinding layers, we show the convolutional layers we analyzed with MFTMA for easier comparison with the MFTMA results. As the reviewer notes, rewinding the FC layers in VGG16 is a good suggestion since we see very little change in these layers across training, perhaps corresponding to the small gradients we see for these layers in Fig. 5. The rewinding experiments are running, and we will update the Si when ready.
>
> -**"Is MFTMA the only method that can examine/verify the above finding?"**
>
> To our knowledge, MFTMA is currently the only method that can capture the relevant manifold geometry for classification. The creation of other/better methods of measuring the representational geometry is an important, though challenging, direction for further work in our view.
>
> -**"Comments. At the first reading, I didn't understand what "restored examples" means, and it took me a while to understand it. The caption for Fig. A.7 has an error; CIFAR100 should be Tiny ImageNet."** Thank you for catching the error! We will correct them in the updated manuscript. We will also add clarifying explanations for 'restored examples' in the updated version.

---

### Official Review · AnonReviewer4 · 2020-10-31
**Nice contribution in understanding generalization and memorization of deep neural networks**

**Rating:** 7
**Confidence:** 3

**Review:**

The paper empirically studies the reason for the phenomenon that deep neural networks can memorize the data labels, even the labels are randomly generated. New geometric measures by replica mean-field theory are applied in the analysis.

The findings of the paper are interesting. It shows the heterogeneity in layers and training stage of the neural net:

i) Memorization occurs in deeper layers; rewinding the final layer to the early weights mitigates memorization.

ii) When memorization happens, the early layer still learn representations that can generalize.

iii) In the training, early activations stabilize first, and deeper layers weights stabilize first.

iv) Near initialization, the gradient is dominated by unpermuted examples.

I have the following questions/comments:

- It is better to further explain the intuition of the Manifold Geometry Metrics.  The current Figure 1(B) is not very clear.

- In Manifold Capacity, what do P and N exactly mean? Is this P the number of classes as used elsewhere?

- The paper explains that by training on permuted examples, the network can learn generalizable representations at the initial training stage because the gradient ignores permuted examples. But why in the later training stage, the early layers and later layers show different generalization properties?

In general, this paper carries well-organized experiments. One shortcoming is that the paper does not provide a methodology to solve the generalization problem or further theoretical analysis of the observations.  But the empirical discoveries are novel and can be beneficial to the deep learning community.

###########

Updates: Thanks for the authors' response. The modified version improves clarity. I think this paper provides nice observations and initial analysis to the community and can be beneficial to future work, so I recommend this paper to be accepted.

---

> ### Author Response · Authors · 2020-11-21
> **Response to Reviewer 4**
>
> We thank the reviewer for their helpful suggestions and are glad that they found our experiments well-organized and findings about the heterogeneity of layers and training stages interesting.
>
> Here we respond to some of the reviewer’s questions/comments:
>
> -**"It is better to further explain the intuition of the Manifold Geometry Metrics. The current Figure 1(B) is not very clear."**
>
> We agree that the intuition behind the Manifold Geometry Metrics should be clarified, including enhancements to the readability and explanation of Fig. 1(B). We have begun to expand these explanations using some of the additional space provided (see the updated document) and will continue to hone this section for clarity.
>
>  -**"In Manifold Capacity, what do P and N exactly mean? Is this P the number of classes as used elsewhere?"**
>
> Here and elsewhere we use P as the number of classes, N as the number of features, and M as the number of examples per class.
>
> -**"The paper explains that training on permuted examples, the network can learn generalizable representations at the initial training stage because the gradient ignores permuted examples. But why in the later training stage, the early layers and later layers show different generalization properties?"**
>
> This is a very interesting question and we do not yet have a good theoretical explanation for this effect. We are so far only able to explain the behavior in the early epochs of training (where permuted examples are largely ignored) but in later epochs we can no longer use the same methods as our linearization about initialization is no longer valid. Additionally, our empirical results on measuring the gradients on unpermuted versus permuted examples suggests that permuted labels contribute strongly to the training in all layers in the later epochs, perhaps owing to the increased nonlinearity of the network after many epochs of training. We speculate that whatever the mechanism that results in the differentiation between early and late layers in the late epochs of training is, it is probably qualitatively different from those that govern the behavior early epochs. We certainly hope that this question can be explored and answered in future works.

---

### Author Response · Authors · 2020-11-21
**Overall response**

We thank all reviewers for their insightful and constructive comments. We appreciate that each reviewer found our approach and findings to be a useful step towards a better understanding of generalization and memorization in DNNs. In the posted revision, we have started incorporating some of the changes suggested by the reviewers. We have expanded the explanation (page 4) and illustrations (Fig 1B) of the replica-theory-based MFTMA method in the main paper. We have also begun incorporating some of the suggested changes on Figures 1, 2, 3, and 5. We will continue to work on the presentation of our figures to improve readability and visual clarity. We have responded to each reviewer's individual questions in our reviewer-specific responses.

---

### Decision · Program_Chairs · 2021-01-07
**Final Decision**

**Decision:**

Accept (Poster)

**Comment:**

The paper offers novel insights about memorization, the process by which deep neural networks are able to learn examples with incorrect labels. The core insight is that late layers are responsible for memorization. The paper presents a thorough examination of this claim from different angles. The experiments involving rewinding late layers are especially innovative.

The reviewers found the insights valuable and voted unanimously for accepting the paper. The sentiment is well summarized by R2: "The findings of the paper are interesting. It shows the heterogeneity in layers and training stage of the neural net".

I would like to bring to your attention the Coherent Gradients paper (see also R1 comment). This and other related papers already discusses the effect of label permutation on the gradient norm. Please make sure you discuss this related work. As a minor comment, please improve the resolution of all figures in the paper.

In summary, it is my pleasure to recommend the acceptance of the paper. Thank you for submitting your work to ICLR, and please make sure you address all remarks of the reviewers in the camera-ready version.